



# Marie Byrd Land glacier change driven by inter-decadal climate-ocean variability

Frazer D.W. Christie[1], Robert G. Bingham[1], Noel Gourmelen[1], Eric J. Steig[2], Rosie R. Bisset[1], Hamish
D. Pritchard[3], Kate Snow[1] and Simon F.B. Tett[1]

[1]School of GeoSciences, University of Edinburgh, Edinburgh, UK
[2]Department of Earth & Space Sciences, University of Washington, Seattle, USA
[3]British Antarctic Survey, Cambridge, UK

*Correspondence to*: Frazer D.W. Christie (F.Christie@ed.ac.uk)

**Abstract.** Over the past 20 years satellite remote sensing has captured significant downwasting of glaciers that drain the West
Antarctic Ice Sheet into the ocean, particularly across the Amundsen Sea Sector. Along the neighbouring Marie Byrd Land
Sector, situated west of Thwaites Glacier to Ross Ice Shelf, glaciological change has been only sparsely monitored. Here, we
use optical satellite imagery to track grounding-line migration along the Marie Byrd Land Sector between 2003 and 2015, and
compare observed changes with ICESat and CryoSat-2-derived surface elevation and thickness change records. During the
observational period, 33% of the grounding line underwent retreat. The greatest retreat rates were observed along the 650-km-
long Getz Ice Shelf, further west of which only minor retreat occurred. The relative glaciological stability west of Getz Ice
Shelf can be attributed to a divergence of the Antarctic Circumpolar Current from the continental-shelf break at 135° W,
coincident with a transition in the morphology of the continental shelf. Along Getz Ice Shelf, grounding-line retreat reduced
by 68% during the CryoSat-2 era relative to earlier observations. This slowdown is a likely response to reduced oceanic forcing,
as inferred from climate reanalysis data. Collectively, our findings underscore the importance of spatial and inter-decadal
variability in climate and ocean interactions in moderating glaciological change around Antarctica.

## 1 Introduction

Recent in situ and satellite remote sensing campaigns have played an important role in constraining the relative roles of ice,
ocean and climate interactions responsible for controlling the substantial ice losses observed in the Amundsen Sea Sector of
West Antarctica over the last ~25 years (Rignot et al., 2008; Mouginot et al., 2014). Comprehending the drivers of these ice
losses is imperative for projecting contributions to global sea-level rise from West Antarctica in the coming decades (e.g.,
Vaughan et al., 2013). Observations of the ice streams and glaciers draining into this sector, in particular Pine Island and
Thwaites Glaciers, have revealed rapid grounding-line retreat (cf. Park et al., 2013; Rignot et al. 2014; Scheuchl et al., 2016),
pronounced ice-dynamic thinning (Pritchard et al., 2009; 2012; Paolo et al., 2015; Konrad et al., 2017), ice-flow-speedup
(Mouginot et al., 2014; Gardner et al., 2017), and large ice-shelf melting rates (Depoorter et al., 2013; Rignot et al., 2013;



Gourmelen et al., 2017). These phenomena have been attributed to oceanic and climatic forcing impinging on the West Antarctic margin (e.g., Jacobs et al., 2011; Dutrieux et al., 2014; Webber et al., 2017).

Whilst the processes driving ice-ocean-climate interactions are being increasingly elucidated for the Amundsen Sea Sector (Jenkins et al., 2016; Asay-Davis et al., 2017; Turner et al., 2017), they remain poorly constrained elsewhere along coastal West Antarctica owing to a dearth of glaciological, oceanographical and climatological observations. Some recent studies have highlighted accelerated dynamic thinning along the Bellingshausen Sea margin (Helm et al., 2014; Paolo et al., 2015; Wouters et al., 2015; Christie et al., 2016; Gardner et al., 2017) potentially linked to climate-ocean forcing similar to that at work in the Amundsen Sea Sector (e.g. Holland et al., 2010; Zhang et al., 2016). Along the Marie Byrd Land coastline of West Antarctica between Thwaites Glacier and the Ross Ice Shelf, little published research exists on the pace and variability of glaciological change or its drivers. Nonetheless, parts of this region have also exhibited rapid downwasting and probable dynamic imbalance over at least the last two decades (e.g., Pritchard et al., 2012; Shepherd et al., 2012; McMillan et al., 2014), notably along the ~650 km-wide Getz Ice Shelf, which has been identified as one of the largest contributors of meltwater originating from sub-ice-shelf melting in Antarctica (Depoorter et al., 2013; Jacobs et al., 2013; Rignot et al., 2013).

In this paper, we present changes in the position of the grounding line along the Marie Byrd Land coastline of West Antarctica, as identified from medium-resolution optical satellite imagery (Landsat and ASTER) between 2003 and 2015. We also recover contemporaneous ice-thinning rates from ICESat laser-altimetry and CryoSat-2 radar-altimetry records. We compare these glaciological observations with ERA-Interim climate reanalysis records of the offshore wind field, which acts as a proxy for the intrusion of warm circumpolar deep water (hereafter CDW) onto the continental shelf.

## 2 Methodology

We define our study domain as the coastline of West Antarctica from parallels 114° W to 157° W (Fig. 1). We term this region, which encompasses the western periphery of Getz Ice Shelf to the eastward limit of the Ross Ice Shelf front, the Marie Byrd Land Sector (hereafter MBLS).

### 2.1 Grounding-line detection and change quantification

To track changes in the position of the MBLS grounding line (hereafter GL), we follow the methodology detailed in Christie et al. (2016). For background, we briefly lay out the main principles of this technique.

We use medium-resolution optical satellite imagery to delineate the break-in-slope across the grounding zone, otherwise known as the "inflexion point", $I_b$, defined as the most seaward continuous surface-slope break detectable in optical satellite imagery (Scambos et al., 2007; Fricker et al., 2009). Used as a proxy for the true GL, which cannot be recovered directly from



satellite remote sensing, $I_b$ appears as a clearly-defined shadow-like change in on-screen pixel intensity (cf. Bindschadler et al., 2011). Most commonly situated approximately 1-2 km downstream of the true GL, $I_b$ typically represents the structural transition from undulating, subglacial-terrain-modulated grounded ice to smoother, floating ice where basal stresses tend towards zero seaward of the GL (see Christie et al., 2016; their Fig. S1).

We used Landsat optical imagery as our primary data source owing to its unmatched, near complete spatial-temporal coverage of the MBLS compared to other freely available remote sensing datasets capable of detecting GL position (cf., Brunt et al., 2011; Rignot et al., 2011). The data were acquired by Landsat's Enhanced Thematic Mapper (hereafter ETM+) and Operational Land Imager (hereafter OLI) sensors, on-board the Landsat 7 and 8 platforms, respectively. In addition, for selected sites during 2003, where Landsat-based mapping was precluded due to persistent cloud cover or poor data quality, we utilised geometrically corrected Terra-ASTER (Advanced Spaceborne Thermal Emission and Reflection Radiometer) Level 1T optical data to supplement our analyses (Fig. S1; Table S1). We tracked the position of $I_b$ along the MBLS at approximately 50-100 m intervals for years 2003, 2008, 2010 and 2015, in order to calculate proxy-GL change over the ICESat (2003-2008) and CryoSat-2 (2010-2015) orbital campaign periods (cf. Pritchard et al., 2009; 2012; Helm et al., 2014). The majority of scenes utilised were acquired during austral summertime (January, February, March), and all scenes had a cloud coverage of <40%. For 2003 and 2008, additional Landsat scenes acquired in December 2002/2007 were utilised when 2003/2008 Landsat and ASTER spatial coverage was restricted by excessive summertime cloud cover. The final $I_b$ products were smoothed using standard GIS tools, and reflect the mean summertime GL position for each year as resolved from all available Landsat or ASTER imagery.

There are certain conditions under which $I_b$ from optical imagery acts as a poor proxy for the true GL. Such circumstances include the mapping of $I_b$ over lightly grounded ice plains, where multiple or no continuous breaks-in-slope are detectable in optical imagery; instances where the location of $I_b$ lies many kilometres landward or seaward of the true GL near ice-plains or ephemerally grounded pinning points (cf. Fricker & Padman, 2006; Fricker et al., 2009; Brunt et al., 2010; Brunt et al., 2011); and over fast-flowing ice streams, where shallow ice-surface slopes, pronounced ice-surface flowlines, and dense crevasse fields render the location of the break-in-slope ambiguous or impossible to delimit (cf. Bindschadler et al., 2011; Rignot et al., 2011). Within the MBLS, such regions include the fast-flowing Berry and De Vicq Glaciers (> 1000 m yr$^{-1}$; Rignot et al. (2011)), as well as a minor number of small-scale suspected ice plains and pinning points throughout the Getz and Nickerson Ice Shelves. We therefore excluded such sites, which account for <1% of the MBLS coastline, from our analysis.

Errors in delineating $I_b$ from optical imagery are derived as a function of satellite orbital geometric error, sensor spatial resolution, and $I_b$ pixel classification confidence, following the protocol outlined in Bindschadler et al. (2011) and Christie et al. (2016; their Text S1 and Table S2). For any given year, we estimate 1σ positional uncertainty to equal approximately ± 100 m (Landsat data) or ± 47 m (ASTER data) along the majority of the MBLS coastline (Table S1).



To propagate error between successive Landsat $I_b$ observations, associated with combining ETM+/ETM+ (for mapped $I_b$ years circa 2003, circa 2008) or ETM+/OLI (years 2010, 2015) sensor data, we additionally calculated the root-sum-square of the 1α-positional uncertainty values calculated for each sensor. This yielded a mean standard error of ~±140 m for most of the

5    main coastline and surrounding islands bounded by ice shelves (Table S2). To calculate uncertainty in 2003-2008 analyses, where 2003 ASTER L1T imagery was utilised in lieu of missing Landsat spatial coverage, the same calculation was applied. Whilst the propagated uncertainty associated with combining ASTER L1T and Landsat ETM+ data (±113m; Table S2) is less than that calculated for combined ETM+/ETM+ and ETM+/OLI sensor observations, for ease of comparability between multi-sensor analyses (Figs. 1 and 2), error values were upscaled to match Landsat-based estimates (~±140 m).

For most of the coastline, any overall imprecision in locating $I_b$ (on the order of $\sim 10^{1-2}$ m) is outweighed by changes in its position over 5 years, i.e. GL advance or retreat on the order of $\geq 10^2$-$10^3$ m. These uncertainties broadly match the positional errors reported in other satellite-based GL detection studies (cf. Brunt et al., 2010; 2011; Rignot et al., 2011; Joughin et al., 2016; Scheuchl et al., 2016). Additional confounding variables such as diurnal tidal variability and atmospheric forcing -

previously recognized as important controls on GL migration over much shorter temporal baselines (Anandakrishnan et al., 2003; Fricker et al., 2009; Brunt et al., 2010) - are assumed to be negligible over the timescales we consider (cf. Milillo et al., 2017).

Upon completion of mapping, GL advance and retreat magnitudes were derived using the procedure detailed in Christie et al.

(2016; their Sect. 2.3 and Text S2). Here, we defined our mapped 2015 $I_b$ line as a baseline. This baseline was partitioned into 30 km segments along the length of the MBLS coast, which permitted the derivation of normal polylines extending infinitely landward and seaward along the outer limits of each segment, intersecting the mapped $I_b$ lines recovered for earlier years. This enabled the creation of GL change polygons representing $I_b$ migration (advance, retreat) over ($2015_{baseline} - y$), where $y$ = earlier mapped year of interest, and permitted the calculation of $I_b$ advance/retreat rates between 2003-2008 and 2010-2015 (Data Set

S2).

## 2.2 Surface elevation and floating ice thickness changes

We compared GL migration along the MBLS with contemporaneous surface elevation change rates derived from ICESat laser-altimetry (2003-2008) and CryoSat-2 radar-altimetry records (2010-2016). We used the data of Pritchard et al. (2009; 2012) to ascertain ICESat era surface elevation change rates (Δh/Δt) over grounded and floating ice within our domain. Following

Pritchard et al. (2009; their Text S1) and Pritchard et al. (2012; their Text S1.1), these datasets were derived from the interpolation of successive near-repeat track median-filtered data acquired over the MBLS, and were converted into smoothed, 10 km grids of mean Δh/Δt calculated over a 30 km radius. Grids of Δh/Δt over grounded and floating ice were processed independently to avoid averaging over the grounding zone, and data acquired over other areas suspected to be not freely-



floating were culled prior to the gridding. Following Pritchard et al. (2012; their Text S1.4 and Table S1), we estimate mean absolute uncertainty in ICESat area-averaged $\Delta h/\Delta t$ to be ~ ± 0.04 m yr$^{-1}$ and ± 0.08 m yr$^{-1}$ over floating and grounded ice, respectively. Over floating ice, $\Delta h/\Delta t$ was subsequently converted into ice-shelf-thickness change rates ($\Delta T/\Delta t$) following a similar methodology to Pritchard et al. (2012), using an assumed ice density of 917 kg m$^{-3}$ (cf. Shepherd et al., 2010; Paolo et

al., 2015). Propagated uncertainties associated with this conversion equal ± 0.40 m yr$^{-1}$.

To obtain surface elevation changes over the CryoSat-2 era, we applied a novel, recently documented swath processing technique (Foresta et al., 2016; Gourmelen et al., 2017) to CryoSat-2 Synthetic Aperture Radar Interferometric (hereafter SARIn) mode data acquired between 2010 and 2016 over the MBLS. Previously applied to other regions of Antarctica (Christie

et al., 2016; Smith et al., 2017; Gourmelen et al., 2017), this technique offers one to two orders of magnitude more elevation measurements than conventional point-of-closest-approach (hereafter POCA) altimetry techniques, thereby maximising spatial coverage and spatial-temporal resolution of ice-sheet marginal areas, including over floating ice (Gourmelen et al., 2017). Numerous corrections were implemented to the L1b dataset during the production of the swath dataset, including, over floating ice, the negation of ocean loading/tidal effects using the CATS2008A tidal model (Padman et al., 2002; following Pritchard et

al., 2012); Gourmelen et al. (2017) provide a comprehensive discussion of these processes.

We derived linear rates of surface elevation change from time-dependent swath elevation between 2010 and 2016 using a plane fit approach on a 10 km grid posting (McMillan et al., 2014). This posting was chosen to match the spatial resolution of our gridded ICESat $\Delta h/\Delta t$ dataset, in order to facilitate comparison of changes in $\Delta h/\Delta t$ between the ICESat and CryoSat-2 eras.

Over floating ice, we additionally employed a Lagrangian framework to derive elevation and rates of surface elevation change to avoid interference associated with the advection of ice-shelf topography through time (Dutrieux et al., 2013; Moholdt et al., 2014; Gourmelen et al., 2017). To do this, we used the MEaSUREs (Making Earth System Data Records for Use in Research Environments) version 2 dataset (Rignot et al., 2017) and additional ice velocity fields generated from feature-tracking of Landsat 8 imagery for the period 2013-2016 (Dehecq et al., 2015) to calculate and assign the position that each CryoSat-2

swath measurement would have had at the beginning of the CryoSat-2 operational period (July 2010). Finally, we corrected for the effects of surface mass balance and firn compaction processes throughout MBLS using the RACMO2.3p2 (5.5 km) and IMAU-FDM (5.5 km) models (Ligtenberg et al., 2011; Lenaerts et al., 2012; Van Wessem et al., 2014; Van Wessem et al., 2016; Gourmelen et al., 2017; Lenaerts et al., 2017). Final uncertainties associated with the swath processing technique were recovered following Gourmelen et al. (2017; their Text S1), and average ± 0.03 m yr$^{-1}$ over both grounded and floating

ice.

Over floating ice, CryoSat-2 era surface elevation change rates were transformed into thickness change rates using the same methodology as for ICESat data, with a propagated uncertainty of ± 0.30 m yr$^{-1}$. Three-monthly shelf-averaged thickness change rates were also obtained as part of the swath processing technique, using the methodology of Foresta et al. (2016), prior



to the Lagrangian correction detailed above. This permitted the comparison of mean shelf-ice thickness changes against the 2003-2008 (ICESat era) and longer-term radar-altimetry-derived record (Paolo et al. (2015), see also Sect. 3).

### 2.3 Ice-ocean-climate proxies

To investigate the role of ocean and climate forcing on glaciological change between 2003 and 2015, we examined mean zonal
wind and Ekman vertical velocity anomalies on and near the MBLS' continental shelf, using ECMWF ERA-Interim climate reanalysis data (cf. Dee et al., 2011).

### 2.3.1 Zonal wind anomalies

Previously established to be a reliable proxy indicator for warm circumpolar deep water upwelling and intrusion onto West Antarctica's continental shelf (Thoma et al., 2008; Steig et al., 2012; Dutrieux et al., 2014), 10 m zonal wind anomalies
(hereafter $U$) along the MBLS continental-shelf slope and break (hereafter CSB) were calculated using ERA-Interim monthly mean of daily mean model outputs (Dee et al., 2011), for all months between January 1979 and December 2016 inclusive. Anomalies were derived by subtracting long-term monthly mean values from the monthly mean of daily mean dataset, beginning in January 1979. To examine inter-decadal variability in $U$, monthly anomalies were averaged over annual timescales to simplify comparison between successive years (see Sect. 3.3).

### 2.3.2 Ekman vertical velocity

A derivative of the wind stress field, Ekman vertical velocity, approximates the rate at which the wind stress curl raises subsurface isopycnals, and can be used as a proxy for Ekman transport-induced upwelling of interior ocean water masses, including relatively warm upper CDW layers (Marshall & Plumb, 2008).

Ekman vertical velocities (hereafter $wE$) were calculated by converting 10 m zonal and meridional wind anomalies into relative wind stresses in fixed grid $x$ (zonal) and $y$ (meridional) planes ($\tau_x$ and $\tau_y$, respectively; Nm s$^{-1}$), using the quadratic stress law detailed in Marshall & Plumb (2008, their Equation 10-1) with an assumed ocean-air neutral drag coefficient of 1.5 x 10$^{-3}$ (dimensionless; cf. MacGregor et al., 2012; Smith et al., 1988). $wE$ was then obtained from:

$$wE = curl((\tau_x, \tau_y) . \frac{1}{p_w f}) \quad \text{(cf. Marshall \& Plumb, 2008; their Equation 10-8)} \qquad (3)$$

$$f = 2\omega Sin(\varphi) \qquad (4)$$

where $f$ denotes variations in the Coriolis parameter with increasing latitude ($\varphi$) and with respect to the Earth's angular velocity ($\omega$, $7.292 \times 10^{-5}$ rad s$^{-1}$); and $p_w$ denotes the assumed density of Ekman layer ocean water ($p_w$; 1027.5 kg m$^{-3}$). $wE$
anomalies were derived from the mean of all ERA-Interim grid cells located on the continental shelf, shelf break and slope of



the MBLS. As for zonal wind calculations (Sect. 2.3.1), monthly anomalies were annually-averaged to examine inter-decadal variability in *wE* (see Sect. 3.3).

**[Fig. 1 here]**

## 3 Results

### 3.1 Grounding-line change 2003-2015

Figure 1 indicates that from 2003 to 2015 GL retreat occurred along ~33% of the MBLS coastline. This quantity incorporates
the entire mainland coastline and Siple and Carney Islands. No net GL advance was observed from 2003 to 2015. The greatest retreat occurred along Getz Ice Shelf, particularly across its central sector between Wright and Dean Islands ('Central' region in Fig. 1, inset), and near the western edge of Getz Ice Shelf, adjacent to De Vicq and Berry Glaciers ('West', Fig. 1 inset). The greatest magnitude of GL retreat totalled $1.72 \pm 0.14$ km on Scott Peninsula (corresponding to Segment 9 in Data Set S2). Locally-averaged GL retreat was $\sim 0.60 \pm 0.14$ km along both the central sector (segments 9-24) and the western edge of the
ice shelf (segments 27-31).

In contrast to Getz Ice Shelf, the GL west of 135° W – which fringes Hull and Land Glaciers and the ice streams draining into Nickerson Ice Shelf – underwent more limited or negligible migration between 2003 and 2015, with maximum retreat occurring across Land Glacier ($0.54 \pm 0.14$ km; Segment 45 in Data Set S2). Farther west, with the exception of minor retreat
near Scambos Glacier ($0.22 \pm 0.14$ km; Segment 83 in Data Set S2), no GL migration took place along the entire perimeter of Sulzberger Ice Shelf, nor throughout the neighbouring Swinbourne Ice Shelf or Bartlett Inlet regions (Fig. 1).

Collectively, our observations highlight the tendency for GL retreat to be clustered around areas of deeply bedded ice (Fig. 1), which correspond to regions of recent moderate-to-fast ice flow inland of the GL ($>500$ m yr$^{-1}$; cf. Rignot et al. (2011)).
Furthermore, partitioning our GL observations into change over the ICESat versus CryoSat-2 campaign periods, we detect several notable patterns of GL migration with respect to ice-sheet altimetry observations. These findings are discussed next.

### 3.2 Glaciological change 2003-2008 (ICESat era)

Between 2003 and 2008, GL retreat along Getz Ice Shelf emulated the spatial pattern observed over the longer timeframe of 2003-2015 (Fig. 1, Fig. 2a). The Getz Ice Shelf over the ICESat era also hosted the largest GL retreat rates observed over the
30 observational period (Fig. 2). Retreat rates along the central and western sectors of Getz Ice Shelf ranged from $34 \pm 30$ m yr$^{-1}$ (Segment 16; Dataset S2) to $319 \pm 30$ m yr$^{-1}$ and $192 \pm 30$ m yr$^{-1}$ on Scott Peninsula and near Berry Glacier, respectively





(Segments 9, 30). Over the same period, no GL migration occurred across the eastern sector of Getz Ice Shelf ('East', inset of Figs. 1 and 2).

The GL retreat observed along the central-western Getz Ice Shelf was also associated with ice surface lowering of up to 120 km inland of the 2008 grounding zone (Fig. 2a). Simultaneously, the ice shelf thinned at an average of $1.76 \pm 0.40$ m yr$^{-1}$ (Fig. 3a), where maximum thinning of $3.8 \pm 0.40$ m yr$^{-1}$ was focussed across the central sector between Scott Peninsula and Carney Island (Fig. 3b), i.e. encompassing the locations at which some of the largest GL retreat rates had occurred. Nevertheless, not all instances of GL retreat over the ICESat era were associated with the locations of highest thinning along Getz Ice Shelf; for example, GL retreat of up to $110 \pm 30$ m yr$^{-1}$ occurred near Dean Island ('D' on Fig. 3(b); Segments 21-23; Data Set S2) where ice-shelf thinning was relatively more limited in magnitude ($\sim$1-1.8 $\pm$ 0.40 m yr$^{-1}$).

West of 135°, almost no GL retreat was observed over the ICESat era outwith Landsat/ASTER 1σ-error bounds (Fig. 2a). In conjunction, ice-surface lowering was less pronounced than along Getz Ice Shelf (Fig. 2a), with negligible overall change in corresponding ice-shelf thinning rates across Nickerson, Sulzberger and Swinbourne Ice Shelves (not shown).

[Fig. 2 here]

### 3.3 Glaciological change 2010-2015 (CryoSat-2 era)

During 2010-2015, GL retreat continued to occur along Getz Ice Shelf but on average 68% slower than in 2003-2008 (Fig. 2b). This reduction in GL retreat was evident even in the central sector of Getz Ice Shelf, where we detect the most rapid GL retreat during this timeframe (Fig. 2b; max. $98 \pm 30$ m yr$^{-1}$; Segment 19, Dataset S2). In addition, a minor segment of the central Getz Ice Shelf GL underwent limited advance ($33 \pm 30$ m yr$^{-1}$; Segment 15, Dataset S2), in conjunction with more minor or negligible GL retreat across both Scott Peninsula and throughout the western sector of the ice shelf (Fig. 2b).

Inland, the ice surface continued to lower (Fig. 2b) as it had done over the previous (ICESat) era (Fig. 2a), with the greatest lowering of grounded ice having occurred upstream of the central and western sectors of the ice shelf. The most notable contrast in Δh/Δt between the two eras was an enhanced thinning signal up to ~50 km inland of the 2015 GL along the central Getz Ice Shelf between Scott Peninsula and Siple Island, and at and near De Vicq and Berry Glaciers, where thinning rates intensified and propagated up to 82 km inland (Fig. 2b; Fig. S2). Over the same period, with the exception of a slowdown in cumulative thickness change between 2011 and 2013, Getz Ice Shelf thinned at an average rate of $1.51 \pm 0.30$ m yr$^{-1}$, not significantly different to the rate over the ICESat era ($1.76 \pm 0.40$ m yr$^{-1}$), nor the longer-term (1994-2012) radar altimetry-based record ($\sim$1.60 $\pm$ 0.30 m yr$^{-1}$; Fig. 3a; Paolo et al. (2015, their Table S2)). At the local scale, along a large proportion of the central Getz Ice Shelf and west of Dean Island (western Getz Ice Shelf), there was a localised slowdown in ice-shelf thinning ($\geq$ 4.00





± 0.30 m yr$^{-1}$) corresponding to the observed locations of slowdown in GL retreat rate (Fig. 3b and c; Fig. S3). A few locations experienced a localised increase in ice-shelf thinning, e.g. west of Berry Glacier, where we observe a reduction in GL retreat, and downstream of De Vicq Glacier, where no GL change information could be recovered from our optical mapping technique (cf. Sect. 2.1.1).

Across the remainder of the MBLS, including the eastern Getz Ice Shelf and the region west of 135° W, we detect negligible GL change between 2010 and 2015, apart from localised retreat at Land Glacier (86 ± 30 m yr$^{-1}$; Fig. 2b; Segment 45 in Dataset S2). With the exception of Hull and Land Glaciers and near Swinbourne Ice Shelf, where surface elevation change rates increased inland of the GL (Fig. 2a and b, Fig. S2), these areas exhibited no significant change in the pace of ice surface

lowering (Fig. 2b; Fig. S2) or ice-shelf thinning (not shown) relative to the earlier ICESat era.

**[Fig. 3 here]**

### 3.4 Ice-ocean-climate interactions: Zonal wind and Ekman vertical velocity

Figure 4 displays annually-averaged 10 m zonal wind ($U$) and Ekman vertical velocity ($wE$) anomalies offshore of the MBLS

(114° W to 157° W). We detect a significant positive (westerly) anomaly in $U$ along the CSB of the MBLS during most of the ICESat era (2003-2008; Fig. 4a), the peak magnitude of which was unprecedented within the ERA-Interim data record (max. MBLS $U$ = 1.15 m s$^{-1}$; 2005). This phenomenon is echoed in the $wE$ record (Fig. 4b), which reveals a predominant net upwelling between 2003 and 2008 (max. $wE$ = 2.94 myr$^{-1}$; 2006). As for $U$, this $wE$ anomaly was also exceptional in the ERA-Interim record, although similar peaks were subsequently recorded in 2009 and 2015. Conversely, for most of the CryoSat-2

era, $U$ was exceptionally negative (easterly), with averaged zonal wind reaching record negative values during 2012 (min. $U$ = -1.30 m s$^{-1}$). This wind anomaly was associated with an overall marked reduction in the tendency for upwelling to occur across MBLS from circa. 2008/2009, with minimum upwelling ($wE$ = -3.34 m yr$^{-1}$) rates also recorded in 2012. Following 2013, $U$ tended back towards zero, becoming positive (westerly) from 2014 onwards, and this was responsible for the significant upwelling event observed in 2015 (max. $wE$ = 3.21 m yr$^{-1}$).

Differences in the spatial distribution of $wE$ for 2010-2014 minus 2003-2008 relative to the long-term ERA-Interim record are plotted in Fig. 5. Comprising of persistently easterly $U$ anomalies prior to 2014 (Fig. 4a), this epoch was characterised by a much reduced $wE$ throughout the MBLS relative to 2003-2008 (Fig. 5). Predominantly centred at ~140-145° W, which ranges spatially between Sulzberger Ice Shelf and the western edge of Getz Ice Shelf, additional, strongly negative Ekman vertical

velocities were also present along the eastern edge and much of the central Getz Ice Shelf during this epoch. Over the same period, a notable reduction in $wE$ near ~115° W, 70° S also occurred.



In summary, the ERA-Interim observational record shows the predominance of Ekman upwelling during 2003-2008, forced by the strongest westerly wind anomaly in the record, followed by a tendency for reduced Ekman upwelling and an easterly wind anomaly during most of the CryoSat-2 era.

**[Fig. 4 here]**

**[Fig. 5 here]**

## 4 Discussion

Our observations have highlighted that over a 12-year period there has been considerable spatial and temporal variability in the nature and pace of GL migration throughout the MBLS (Figs. 1 and 2). In the following sections, we examine the potential
drivers of these phenomena, with respect to regionally contrasting ice, ocean, climate and other potential interactions at work across the MBLS. To support our discussion we make reference to a conceptual model presented in Fig. 6.

**[Fig. 6 here]**

### 4.1 Getz Ice Shelf grounding-line change 2003-2015

The most prominent GL retreat throughout the MBLS occurred along the ~650 km Getz Ice Shelf (Fig. 1; Sect. 3.1), which neighbours the recent, rapidly downwasting ice-masses of the wider Amundsen Sea Sector. This was a likely consequence of the substantial thinning and basal melting witnessed over this region in recent decades (Figs. 2 and 3; see also Pritchard et al., 2009; 2012; Jacobs et al., 2013; Paolo et al., 2015). All the GL retreat within the central and western sectors of Getz Ice Shelf occurred directly upstream of well-surveyed, deep (>400 m) bathymetric depressions north of the ice fronts (Fig. 1), which
transect the continental shelf and route warm modified-CDW from the CSB to the sub-ice shelf cavity (Wåhlin et al., 2010; Arneborg et al., 2012; Jacobs et al., 2013; see also our Fig. 6a). Aping the offshore geological setting of the Bellingshausen and eastern Amundsen Sea Sectors, where recent glaciological change is believed to have been facilitated by CDW ingress along similar cross-continental-shelf conduits (Nitsche et al., 2007; Walker et al., 2007; Bingham et al., 2012; Pritchard et al., 2012; St-Laurent et al., 2013; Zhang et al., 2016), these observations are strongly suggestive of ocean-forced glaciological
change at work across the central-western Getz Ice Shelf and its inland basins between at least 2003 and 2015.

Along the eastern sector of the ice shelf, where we detect almost no GL migration between 2003 and 2015 (See Sect 3.1. and 3.2.; Fig. 1), ice-flow velocities (Rignot et al., 2011; Gardner et al., 2017) and ice-thinning rates (Figs. 2 and 3) were more limited relative to the remainder of the ice shelf. This is despite the proximity of the deep Getz-Dotson Trough to the eastern
sector's ice front (Fig. 1). Collectively, these observations imply that local-scale ice-ocean processes or geological



configurations underneath the most easterly portion of Getz Ice Shelf may render the region relatively immune to ocean-forced dynamic thinning and subsequent GL retreat.

## 4.2 Controls on Getz Ice Shelf grounding-line dynamics

### 4.2.1 Climate-ocean forcing

That GL retreat slowed down consistently along the length of the (~650 km long) central-western Getz Ice Shelf in the CryoSat-2 era relative to the ICESat era (Fig. 2; Fig. S3; Sect. 3.3) is indicative of regional-scale external forcing, primarily with respect to the intensity of ocean and climate forcing of Getz Ice Shelf through time. Indeed, the changes in 10 m zonal wind, $U$ (Fig. 4a), and Ekman vertical velocity, $wE$ (Fig. 4b), suggest that the 2010-2015 era was characterised by a predominantly easterly wind anomaly over the MBLS CSB (Sect. 3.4), in conjunction with an implied regional-scale reduction in CDW upwelling

and flooding onto the continental shelf. These findings are fully synchronous with similar oceanographical-climatological trends observed over the wider Amundsen Sea Sector since 2009 (Dutrieux et al., 2014), which were responsible for a much deepened CDW layer across this region of West Antarctica relative to the ICESat era (cf. Wåhlin et al., 2010; Jacobs et al., 2013; Dutrieux et al., 2014; Webber et al., 2017). Dutrieux et al. (2014; following Steig et al., 2012) attributed this behaviour to inter-decadal variability in tropical-pacific ENSO forcing and so, by extension, Getz Ice Shelf and the wider MBLS also

appear responsive to inter-decadal variability in global-scale climatic forcing. This hypothesis concurs with the earlier findings of Jacobs et al. (2013), who attributed a reduced thermocline and glaciological forcing along Getz Ice Shelf in the years preceding the ICESat era to a strong La Niña event circa. 2000.

Locally, the significant negative difference in $wE$ near the eastern Getz Ice Shelf is also notable (Fig. 5). Widespread along the

eastern and parts of the central Getz coast between 2010-2014 (Fig. 5), this anomaly was prevalent across the deeply-bedded coastal tributaries of the Getz-Dotson Trough (Fig. 1), which act as the primary CDW ingress pathways to the eastern and central sectors of the Getz Ice Shelf (Wåhlin et al., 2010; Arneborg et al., 2012; Jacobs et al., 2013). Likely related to changes in the spatial extent of the Amundsen Sea Polynya since 2003-2008 (cf. Nihashi & Ohshima, 2015; 2017; Kim et al., 2017), we hypothesise that the net reduced upwelling inferred to have occurred at this coastal location, consistent with a much

deepened isopycnal depth over the trough's tributaries, may have been responsible for an additional, locally-forced reduction in central and eastern Getz-bound CDW inflow during 2010-2015. Such a reduction – together with observations of deeply reduced $wE$ along much of the remaining Getz Ice Shelf and wider MBLS between 2010 and 2014 (Fig. 5)- may explain the overall, central-western Getz-wide reduction in GL retreat relative to the 2003-2008 (ICESat) era, and may have simultaneously bolstered the eastern ice shelf's overall immunity to pervasive ocean-forced GL retreat (Figs. 1 and 2). Similar

processes are believed to have governed the magnitude of change at Pine Island Glacier in recent years, where a dramatic reduction in basal melt rate and other glaciological change was observed between 2011 and 2013 (Dutrieux et al., 2014; Mouginot et al., 2014; St. Laurent et al., 2016; Webber et al., 2017). Over the same timeframe, where we detect minimum $U$

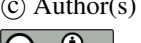



and *wE* across the MBLS (Figs. 4a and b; Sect. 3.4), such processes may also have induced the temporary hiatus in shelf-averaged thickness change witnessed between 2011 and 2013, where mean Getz Ice Shelf thinning rates appear to have abated contrary to the longer-term CryoSat-2 trend (Fig. 3a; also Sect. 3.3).

Overall, our findings underscore the importance of inter-decadal variability in both regional- and local-scale climate and ocean interactions in moderating glaciological change along this sector of Antarctica (Fig. 6). However, whilst an apparent correlation exists between Getz Ice Shelf GL retreat rate and inter-decadal variability in climate-ocean forcing (cf. Sect. 3.4 and 4.2.1), observations of generally unabated CryoSat-2 shelf-averaged thinning rates relative to earlier records (Fig. 3a; Pritchard et al. (2012); Paolo et al. (2015)) suggest that additional processes may have facilitated the observed slowdown in 2010-2015 GL

retreat along parts of ice shelf. That is, such temporally sustained ice-shelf thinning (Fig. 3a) would be expected to augment progressive GL retreat over time (cf. Schoof, 2007), contrary to the observed GL-retreat slowdown. Possible confounding factors are surface mass balance processes, grounding zone bed geometry, and local-scale changes in ice dynamics.

### 4.2.2 Surface mass balance processes

Temporal changes in surface mass balance must be accounted for in order to resolve the relative influence of surface- and/or

dynamically-induced (i.e. ocean-forced) processes controlling surface elevation change (see Pritchard et al., 2012; Helm et al., 2014; Wouters et al., 2015 for further discussion). For example, marked inter-annual variability in surface accumulation was believed to have played a leading role in driving the enhanced surface lowering witnessed over the Southern Antarctic Peninsula in recent years (cf. Helm et al., 2014; following Thomas et al., 2008).

Near Getz Ice Shelf, where several studies have highlighted trends of increased ice mass loss in recent years (e.g., McMillan et al., 2014; Gardner et al., 2017), negative trends in surface mass balance have recently been documented inland of the GL (up to 0.80 m yr$^{-1}$ between 2006 and 2015; Chuter et al. (2017); see also their Fig. S4). However, whilst negative surface mass balance is believed to have controlled a large proportion of ice loss over grounded ice in the MBLS (Chuter et al., 2017), we note that that such decreases are insufficient to explain the pronounced ice-shelf thinning and basal melting rates observed

seaward of the GL over the ICESat (Fig. 3a; Depoorter et al., 2013; Jacobs et al., 2013) and CryoSat-2 (Sect. 3.3 and Fig. 3a) eras. Moreover, recent modelled estimates over Getz Ice Shelf reveal only minor temporal changes in surface mass balance trend (Lenaerts et al., 2016), with 2010-2015 annual mean rates of surface mass balance (0.8 m w.e. yr$^{-1}$; Figure S4) not significantly different to the long term (1979-2015) average (~ 0.8-1.0 m w.e. yr$^{-1}$; cf. Lenaerts et al., 2017; their Figure 7). By implication, surface mass balance changes are therefore unlikely to have played a significant role in controlling the progressive

thinning of Getz Ice Shelf over the CryoSat-2 era (Fig. 3a), or the observed slowdown in GL retreat rate (Figs. 2a and b; Fig. S3).




### 4.2.3 Grounding zone bed geometry

Bed conditions at the grounding zone of Getz Ice Shelf are poorly constrained, and are predominantly interpolated from a limited number of Operation IceBridge radar depth-sounding data and other lower-resolution geophysical datasets (Fretwell et al., 2013). Nonetheless, we note that upstream of the 2008 GL position, the inclination of surface and bed slopes along a large proportion of the Getz Ice Shelf, as inferred from Bedmap2 (Fretwell et al., 2013) and optical satellite imagery (Data Set S1), become increasingly prograde and undulating, which may act to inhibit GL retreat. That is, the potentially rough and shoaling topography upstream of the 2008 GL may require prolonged, continuously high, or increased rates of thinning to permit GL retreat – a process that has been modelled across both idealised and physically-constrained grounding zone geometries over other parts of West Antarctica (cf. Schoof et al., 2007; Durand et al., 2011; Parizek et al., 2013; Nias et al., 2016). Until more comprehensive knowledge of Getz Ice Shelf's grounding zone bed structure exists, glacier/ice-stream-specific internal variability, moderated by bed conditions at the 2010-2015 grounding zone, cannot be reliably dismissed as an additional control on the slowdown of GL retreat rate during the CryoSat-2 era. Increased geophysical survey of the entire grounding zone is therefore an important scientific objective towards more accurately assessing the future evolution of Getz Ice Shelf GL migration and glaciological (in)stability in the coming decades.

### 4.2.4 Changes in ice dynamics

Changes in ice dynamics, potentially linked to differences in subglacial bed conditions, may also have played an important role in influencing GL retreat slowdown along Getz Ice Shelf during the observational period. Gardner et al. (2017) have presented changes to ice surface velocities across this region between circa. 2007/2008 and 2015. There is strong correspondence along the Getz Ice Shelf margin between the locations where we have observed GL slowdown or speedup (Fig. 2; Fig. S3) with those where Gardner et al. (2017) documented ice velocity slowdown or speedup over the same time interval. Locations of ice-velocity speedup are localised along the coastline but represent the major contributors to a recorded 6% increase in ice-mass discharge across the MBLS (Gardner et al., 2017). These locations, including the fast-flowing Berry and De Vicq Glaciers (Figs. 1 and 2), are likely outlets of thick ice (cf. Jacobs et al., 2013) characterised by a transition towards dynamically-unstable glaciological change that is now divorced from climate-ocean forcing. Everywhere else along the central-western Getz Ice Shelf GL, the ice flow remained constant or decelerated (Gardner et al., 2017; their Fig. 8, panel 20), which is consistent with the ice-shelf-wide reduction in climate-ocean forcing noted in Sect. 4.2.1.

### 4.3 Grounding-line change and its controls west of Getz Ice Shelf

Along the coastline between Getz and Ross Ice Shelves, encompassing the GL along Nickerson and Sulzberger Ice Shelves, GL migration was negligible or exhibited lower retreat than along Getz Ice Shelf between 2003 and 2015 (Fig. 1; Sect. 3.1). In addition, ice-surface lowering along the majority of this coastline was negligible over the observation window (Fig. 2).



One explanation for the muted GL response west of Getz Ice Shelf may be provided by the region's underlying geology. Along this coastline prominent mountains, likely to be of volcanic origin (Fretwell et al., 2013; Van Wyk de Vries et al., 2017), rise steeply immediately inland of the grounding zone, providing a possible regional topographic barrier to pervasive GL retreat and dynamic thinning. The few locations of notable GL retreat west of Getz Ice Shelf, for example at Land and Scambos

Glaciers (Fig. 1), may be attributed to breaches of this coastal topographic barrier. These locations reside immediately upstream of deep (> 500 m) subglacial depressions which extend to within close proximity of the continental-shelf margin (Fig. 1; S5; Arndt et al., 2013), and which likely represent the routeways of warm-based, fast-flowing ice-streams during glacial maximum conditions (Ó Cofaigh et al., 2005; Nitsche et al., 2016).

Another inhibitor of GL change west of 135° W may derive from regional contrasts in the structure of the continental shelf seaward of Getz Ice Shelf versus farther west. In neighbouring regions of West Antarctica, continental shelves fronted by shallower shelf slopes and bisected by multiple troughs are critical for permitting sustained upwelling, ingress and shoreward transportation of CDW towards the GL (e.g., Bellingshausen Sea Sector (Ó Cofaigh et al., 2005; Bingham et al., 2012), Amundsen Sea Sector including Getz-Dotson Trough (Walker et al., 2007; Wåhlin et al. 2010)). By contrast, the continental

shelf west of 135° W is characterised by a shallower seafloor fronted by a steeper continental shelf slope, and the general absence of significant, shelf-bisecting troughs that breach the CSB (Fig. 1; Fig. S5). Indeed, from in-situ measurements Jacobs et al. (2013) recovered only intermittent CDW presence along the length of the CSB west of 135° W, while Schmidtko et al. (2014) showed that rates of long-term CDW shoaling across the same region were more restricted compared with the Amundsen and Bellingshausen Sea Sectors. Recent work utilising an ocean eddy-resolving model, parameterised to account

for the effects of variable continental-shelf geometry on CDW ingress, has also underscored that reduced CDW transport onto the continental shelf can be attributed to steep continental slope bathymetry in conjunction with a more pronounced Antarctic Slope Front over the CSB (Stewart & Thomson, 2015; their Fig. 2d). The Antarctic Slope Front acts to separate fresh continental-shelf surface waters from offshore CDW sources (Jacobs, 1991; Baines, 2009; Stewart & Thomson, 2015, their Figs. 1b and 2d). Together, these findings suggest that the strength of the Antarctic Slope Front may be critical to the

vulnerability of the MBLS to CDW-induced GL retreat and dynamic instability (Fig. 6). Notably, Whitworth et al. (1998) traced the beginnings of a steeper Antarctic Slope Front extending westward from 120° W, while Lee & Coward (2003; cf. Thompson, 2008) independently inferred a transition to a steeper Antarctic Slope Front from modelled ocean surface current velocities at ~125-135° W. These longitudinal limits correspond broadly with the contrasting GL behaviours we have observed along and west of Getz Ice Shelf (Figs. 1 and 2).

It is important to note that the strength of the Antarctic Slope Front co-exists with, and is speculated to be influenced by, the positioning of the Antarctic Circumpolar Current (hereafter ACC) (Walker et al., 2013). The ACC drives circumpolar westerly ocean circulation and ultimately influences the upwelling and delivery of CDW from the deep ocean towards the continental shelf (Orsi et al., 1995). Across the MBLS, we note that the steepening of the Antarctic Slope Front at ~135° W strongly aligns





with a marked northward deflection of the ACC from the continental-shelf slope limits at ~130-135° W. This ACC behaviour persists from here to the western limits of the Ross Sea at ~150° E (Fig. S6; cf. Orsi et al. (1995)), and likely partly explains the limited presence of CDW west of 135° W (cf. Jacobs et al., 2013; see also our Fig. 6b). In contrast, east of 135°W, the ACC more closely follows the shallower continental-shelf slope margins adjacent to the Getz Ice Shelf and Amundsen and

Bellingshausen Sectors (Fig. S6), contributing to the previously documented presence of on-shelf CDW across these regions (Holland et al., 2010; Jacobs et al., 2013; Zhang et al., 2016) and, most likely, the observed patterns of ocean-driven GL retreat and sustained thinning throughout the Getz Ice Shelf between 2003 and 2015 (Fig. 6a).

Finally, we acknowledge that the large meltwater fluxes originating from the Amundsen Sea Sector in recent years (Jacobs et
al., 2013; Rignot et al., 2013; Depoorter et al., 2013) may also play an important role in moderating ice-ocean interactions west of 135° W. A recent modelling study suggests that up to one third of the total meltwater derived from the Amundsen Sea Sector is transported towards the Ross Sea (Nakayama et al., 2014). Within this trend, up to 50% of the total meltwater content delivered to the Ross Sea originates from Getz Ice Shelf via a pronounced easterly coastal current, and Ross-Sea-bound meltwater-transportation pathways flood the entire width of the continental shelf west of 135° W (Nakayama et al., 2014; their
Figs. 2b and 3c). This implies that the high volumes of meltwater originating from Getz Ice Shelf in recent years (Rignot et al., 2013), even during steady-state conditions, may be sufficient to result in the enhanced modification of on-shelf CDW across this region of the MBLS. Hypothesised to become exacerbated by future increases in dynamic basal melting of Amundsen Sea ice shelves (Nakayama et al., 2014), this process may continue to limit CDW access to the sub-shelf cavity in the coming decades, and partly explain our observations of near-negligible GL retreat and thinning rates between 2003 and
2015.

## 5 Conclusions

Medium-resolution optical satellite imagery show that ~33% of the Marie Byrd Land coastline, feeding the western Amundsen Sea and eastern Ross Sea, West Antarctica, experienced grounding line retreat between 2003 and 2015. Since 2003, the grounding line has retreated pervasively along Getz Ice Shelf, but farther west has remained predominantly stable. Between
2003 and 2008 (ICESat era) the grounding line retreated more rapidly along the Getz Ice Shelf margin than between 2010 and 2015 (CryoSat-2 era).

We attribute the observed slowdown in Getz Ice Shelf's grounding-line retreat to a reduction in external climate-ocean forcing as inferred from climate reanalysis data. During the CryoSat-2 era, weaker offshore winds relative to the ICESat era reduced
Ekman upwelling on and around the continental shelf, resulting in a decline in Circumpolar Deep Water intrusion to the sub-Getz ice-shelf cavity. This behaviour is analogous to that reported from the wider Amundsen Sea Sector since 2009 (Dutrieux et al., 2014, following Steig et al., 2012; Turner et al., 2017), and mirrors similar trends inferred to have occurred near Getz

Ice Shelf in the years immediately preceding the ICESat era (Jacobs et al., 2013). At the local scale, coastal ice-ocean-climate interactions, bed geometry and changes to the ice-dynamic regime may also have influenced grounding-line retreat rates.

We ascribe the relative glaciological stability of the Marie Byrd Land margin west of Getz Ice Shelf, encompassing Nickerson and Sulzberger Ice Shelves, to a divergence of the Antarctic Circumpolar Current from the continental-shelf break at ~130-135°W. At this longitude, the Antarctic Slope Front also intensifies in conjunction with a steepening of the continental-shelf slope and a shallowing of the continental-shelf floor. In consequence, much of the Antarctic margin between the Getz and Ross Ice Shelves is buffered from the climate-ocean-driven forcing that has been observed farther west.

Collectively, our findings from the Marie Byrd Land Sector underscore the importance of both spatial and inter-decadal variability in ocean and climate interactions for moderating glaciological change around Antarctica.

**Data availability**

All GL, ice frontal position and MBLS ice-mask datasets derived from this study are available at https://doi.org/10.1594/PANGAEA.884782. Landsat data used in this study are available from the USGS/NASA at earth-explorer.usgs.gov/, Cryosat-2 data are available from the European Space Agency, and ERA-Interim data are available from the European Centre for Medium-Range Weather Forecasts (ECMWF) at https://www.ecmwf.int/en/research/climate-reanalysis/era-interim. Supplementary data supporting the results of this paper are available in the accompanying supporting information file.

**Author contributions**

FDWC designed the study, performed all analyses, and wrote the paper under the guidance of RGB. FDWC developed the optically-derived GL detection technique and carried out GL change analysis with the assistance of RRB. NG generated the CryoSat-2 surface elevation change (Δh/Δt) data. EJS gave guidance/support to FDWC in sourcing, deriving and interpreting the ERA-Interim datasets presented in this study. HDP provided the ICESat altimetry data; and KS and SFBT added significant value to the discussion on ice-ocean-climate interactions around West Antarctica, via several in-depth discussions with FDWC.

**Competing interests**

The authors declare that they have no conflict of interest.



**Acknowledgements**

FDWC was funded by a Carnegie Trust for the Universities of Scotland Carnegie PhD Scholarship with RGB, hosted in the Edinburgh E³ U.K. Natural Environment Research Council Doctoral Training Partnership (NE/L002558/1) and the Scottish Alliance for Geoscience, Environment and Society (SAGES) Graduate School. FDWC was also funded through the generous

5   support of a Trans-Antarctic Association Small Grant (TAA17-01) and a SAGES Post-doctoral Early Career Research Exchange (PECRE) award which he used to visit EJS at the University of Washington's Department of Earth and Space Sciences.  NG was funded under the European Space Agency's Support To Science Element CryoTop 4000107394/12/I-NB and CryoTop evolution 4000116874/16/I-NB studies. The authors also wish to thank F. Paolo & H. Fricker for kindly sharing the ERS-1/2 and ENVISAT altimetry data used to generate Fig. 3a, J. Lenaerts, S. Ligtenberg, and W. van de Berg for sharing

10   their RACMO2 and IMAU-FDM models used in the CryoSat-2 swath processing chain, and J. Williams and S. Pinson for their assistance in deriving Landsat 8 velocity fields over the MBLS.





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





**Figure 1.** 2003-2015 net grounding line (GL) change along the Marie Byrd Land Sector of West Antarctica. Circle radii denote the magnitude of GL change per 30 km segment along the MBL grounding line. Small black circles denote negligible GL change detected within satellite error bounds. Note the non-linear scaling of change. Change symbols are overlaid upon MOA2009 grounded and floating ice shelf boundary masks (Haran et al., 2014); as well as IBCSO v.1 circum-Antarctic bathymetry data (Arndt et al., 2013) with 500 m (light grey) and 1000 m (dark grey) depth contours. *Dots*, *Getz*, *Nick*, *Sulz* and *Sw* denote the Dotson, Getz, Nickerson, Sulzberger and Swinbourne Ice Shelves (respectively); *GDT*, Getz-Dotson Trough; *M*, Martin Peninsula; *B*, Bear Island; *W*, Wright Island; *SP*, Scott Peninsula; *DI*, Duncan Island; *C*, Carney Island; *S*, Siple Island; *D*, Dean Island; *DV*, De Vicq Glacier; *G*, Grant Island; *B*, Berry Glacier; *H*, Hall Glacier; *L*, Land Glacier; *Sc*,



Scambos Glacier and *Ba*, Bartlett Inlet (Swithinbank et al., 2003a; 2003b). Note the presence of steep continental slope break gradients situated to the west of the 135° W. North of Dotson and Getz Ice Shelves, also note the deep glacially scoured troughs that transect the continental shelf and connect present day ice fronts to the shelf break (see main text for further discussion). Inset map (top left) = location of the MBLS. Inset (top right) = eastern, central and western sectors of Getz Ice Shelf, as referred to in the main text.





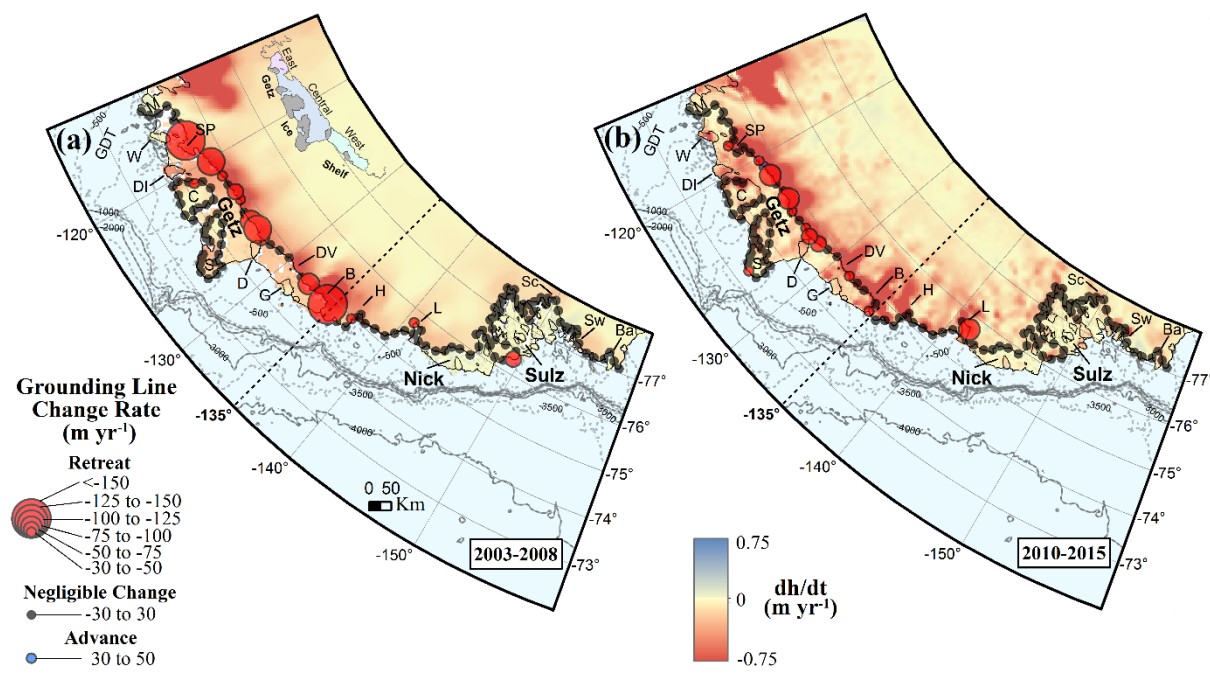

**Figure 2**. Rate of change in GL position throughout the Marie Byrd Land Sector over the periods (*a*) 2003-2008 and (*b*) 2010-2015. Circle radii denote the magnitude and direction of grounding line migration (red, retreat; blue, advance) per 30 km segment across the domain. As for Fig. 1, black circles denote negligible change detected within satellite error bounds. Note the non-linear scaling of change. GL migration data are superimposed over gridded surface elevation change rates ($\Delta h/\Delta t$; m yr$^{-1}$), as derived from (*a*) ICESat (Pritchard et al., 2009; 2012) and (*b*) swath processed Cryosat-2 data (this study). Bathymetric contours, site labels and Getz Ice Shelf inset same as Fig. 1.



**Figure 3**. (*a*) Time-series of cumulative thickness change for Getz Ice Shelf, 1994-2016. Grey squares correspond to shelf-wide, 3-month-average thickness changes relative to the series mean, derived from ERS-1/2, ENVISAT (light grey squares; cf. Paolo et al. (2015)), and CryoSat-2 data (dark grey squares; this study). Black curve denotes the polynomial trend for the entire observational period (1994-2016), superimposed over the 1994-2012 trend (green line) reported in Paolo et al. (2015; their Fig. S1). Blue and red lines denote linear trends over the ICESat and CryoSat-2 eras, respectively. Average rates of





thickness change (m yr⁻¹) were approximated from the derivative of the polynomial fit with respect to time. Whilst thinning rates during the CryoSat-2 era have not differed significantly from the ICESat and Paolo long-term trend, note the apparent hiatus in thickness change between 2011 and 2013, as discussed in Sect. 3.3 and 4.2.1. (*b*) and (*c*) Spatial distribution of Getz Ice Shelf thickness change rates ($\Delta T \Delta t$; m yr⁻¹) over the ICESat and CryoSat-2 eras, respectively. For reference, GL change data from Fig. 2 are also shown. *SI* denotes Shepherd Island; all other site labels and bathymetric contours same as Figs. 1 and 2.





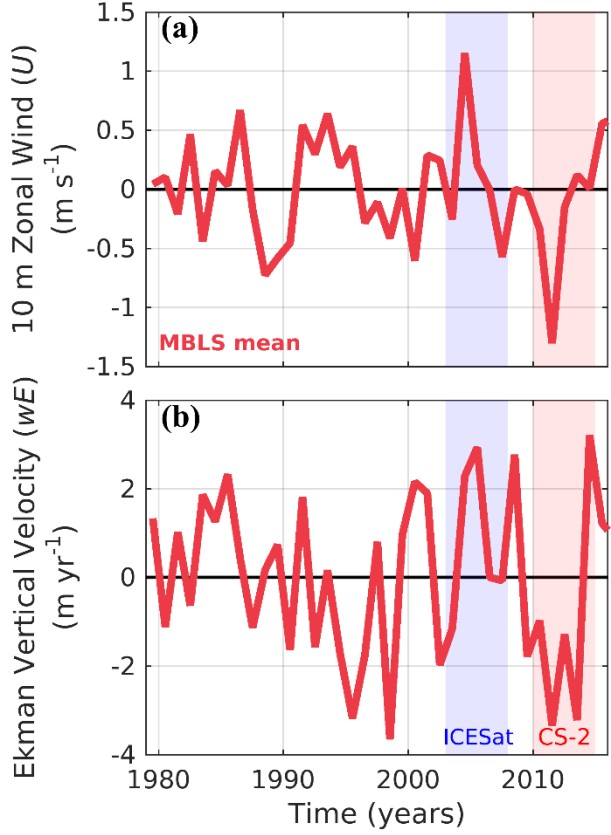

**Figure 4**. (*a*) Mean 1979-2016 10 m zonal wind (*U*) and (*b*) Ekman vertical velocity (*wE*) anomalies over the Marie Byrd Land Sector, derived from ERA-Interim climate reanalysis data. The blue and red patches denote the ICESat and CryoSat-2 eras (Figs. 2a and b). *wE* anomalies were derived from the mean of all ERA-Interim grid cells located on the continental shelf, shelf break and shelf slope. *U* anomalies were averaged along the length of the continental shelf break and slope only (see also Fig. 6). In (*a*), positive values denote anomalous westerly 10 m surface winds; negative, easterly. In (*b*), positive values denote anomalous upwelling associated with Ekman Suction, and negative values denote reduced upwelling by Ekman Pumping. In (*a*) and (*b*), note the unprecedented MBLS *U* and positive *wE* over the ICESat era, compared with the strong negative anomalies during most of the CryoSat-2 era.



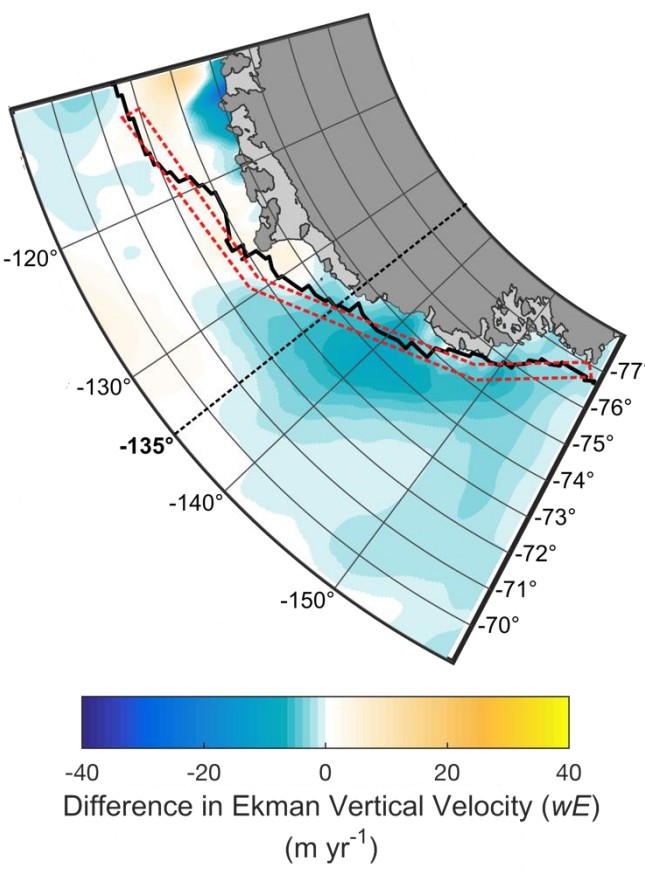

**Figure 5.** Difference between 2010-2014 and 2003-2008 Ekman vertical velocities ($wE$) relative to all preceding years within the ERA-Interim record. Negative values denote reduced Ekman upwelling. The thick black line denotes the approximate location of the continental shelf break at 1000 m depth (Arndt et al., 2013). The red dashed box denotes the region used to derive the mean MBLS 10 m zonal wind anomalies observed in Fig. 4a, and demarcates the northern-most limits of the grid used to derive the MBLS shelf-averaged $wE$ anomalies shown in Fig. 4b. Note the presence of deeply reduced upwelling near the ice-fronts of Dotson and Getz Ice Shelves at ~115° W, in addition to a similar phenomenon west of ~130° W which extends from the coastline to at least ~70° S. Opposite Getz Ice Shelf, also note the reduced $wE$ near ~115° W, 70° S.





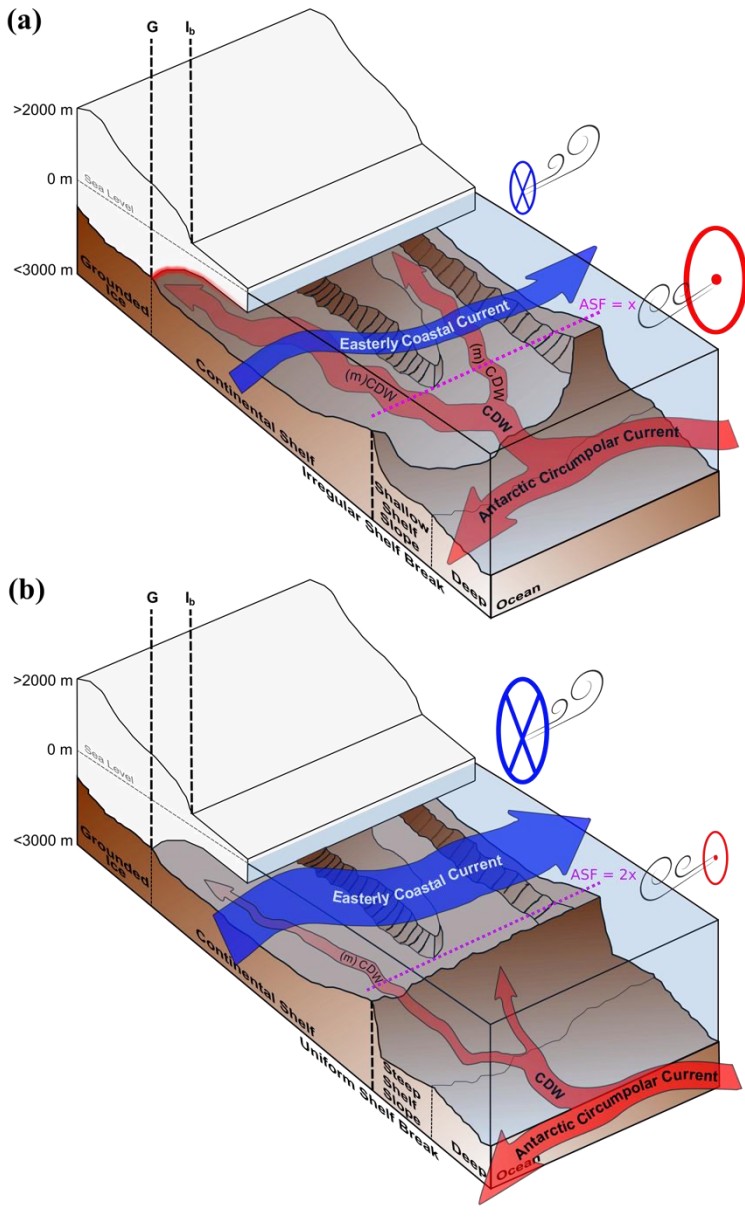

**Figure 6**. Schematic summary of the ocean, climate and geological controls influencing glaciological change along (*a*) Getz Ice Shelf and (*b*) the region west of 135° W. In (*a*), the deep troughs bisecting the continental shelf break allow circumpolar deep water (CDW) to intrude onto the continental shelf and reach sub-ice-shelf cavities as modified CDW, enabling ocean-driven melting of ice and grounding line retreat. CDW is sourced from the Antarctic Circumpolar Current (ACC), located within close proximity to the shelf slope, and is transported upslope via surface wind-driven Ekman Suction, induced by anomalous westerly winds over the shelf break (see main text for further discussion). In (*b*), the steep continental shelf slope





and shallow shelf break result in negligible or only minor access of CDW onto the continental shelf, associated with reduced eddy-mediated transport of CDW over the CSB, in conjunction with a stronger Antarctic Slope Front (ASF) than in (*a*) (Stewart & Thompson, 2015). The northward deflection of the ACC also minimises the presence of CDW near the shelf slope. Relative to (*a*), the strong easterly coastal current, comprising fresh Antarctic surface water (including Ross Sea-bound melt waters from Getz Ice Shelf and the wider Amundsen Sea Sector), acts to freshen the continental shelf water column (cf. Nakayama et al. 2014), resulting in buffered modified CDW access to the sub-ice shelf cavity. $G$ and $I_b$ refer to the true grounding line and inflexion point, respectively.