# Peer review of "Glacier change along West Antarctica's Marie Byrd Land Sector and links to inter-decadal atmosphere-ocean variability"

_The Cryosphere, 2017_

## Referee Comment (RC1) · Anonymous Referee #1 · 19 Feb 2018

The authors analyzed the glacier changes in Marie Byrd Land sector for the past 20 years. Using ICESat and CryoSat-2, they show that grounding line retreat reduced by 68% in CryoSat-2 era, which is caused by oceanic forcing. Although slowdown of grounding line retreat is an interesting and important finding, their argument that observed changes are caused by "reduced Ekman upwelling on and around the continental shelf" is not well supported. I recommend major revision. Very nice results, but some interpretations seem to me rather too speculative. More analyses and/or different interpretations are required.

Major comment :

**1 Very good job listing oceanic processes, which may impact the glacier retreat. Although authors are aware of many oceanic processes, authors conclude that "during**

[Figure]

weaker offshore winds relative to the ICESat era reduced Ekman upwelling on and around the continental shelf, resulting in a decline in Circumpolar Deep Water intrusion to the sub Getz ice-shelf cavity", which is not supported from any of the analyses conducted in the paper.

As authors are aware, oceanic conditions (e.g.,large and small scale circulations, bathymetry, stratification, etc) are very much different in the Marie Byrd Land sector. Since there are many processes potentially controlling potential temperature in the ice shelf cavity and thus ice shelf melt rates and relative importance of these processes are likely regionally different, authors are not able to conclude that " reduced Ekman upwelling on and around the continental shelf" is the key process for this region, just based on the fact that they observe changes in Ekman upwelling. Cited papers such as Steig et al., 2012, Dutrieux 2014, and St. Laurent et al., 2016 have conducted data analysis and/or modeling. Further analysis including data analysis and modeling is likely required to claim that "reduced Ekman upwelling on and around the continental shelf" is the reason for the observed changes.

**2 The title of this paper indicates that glacier change is driven by inter-decadal climate ocean variability, which is misleading. Authors do not show that the impact of other processes are small. There are other processes impacting glacier retreat (section 4.2) and these processes may possibly be more important (e.g., subsection 4.2.3).**

Minor comment

Page 9 Lines 14-25 : It is clearer if authors can show spatial pattern of vertical Ekman velocity for each era (not just the difference as in Figure 5).

Page 11 Lines 10-11 : Even if it is fully synchronous, it is not convincing that "reduced Ekman upwelling on and around the continental shelf" changes the oceanic condition in the ice shelf cavity, reduces the melt rates, and slows down the grounding line retreat. As stated above, there are many processes and further analyses are required.

Page 11 Lines 23-27 : Where do you mean ? Is there different polynya in near Getz region ? If so, are these responding similarly to the Amundsen Sea polynya?

Page 14 Lines 10-29 : "These longitudinal limits corresponds broadly with . . . Getz Ice Shelf". These argument seems speculative. Need more clarification.

Page 15 Lines 22-Page 16 Line 12 : See major comments #1 and #2.

Figure 6 : This seems to be not accurate. Should circulation off the Marie Byrd Land sector be influenced by the Ross Gyre and CDW circulation be opposite ?

---

## Referee Comment (RC2) · Anonymous Referee #2 · 23 Feb 2018

Summary

The authors present an analysis of changes in grounding line position for the Marie Byrd Land Sector using a well established method of detecting the break in slope from satellite remote sensing imagery (Landsat & ASTER). Over the 2003 - 2015 period they find that 33% of the grounding line underwent retreat, with the greatest rates found over the Getz Ice Shelf grounding line region. These results are consistent with the ice shelf thinning rates presented in the paper and previous studies in the literature. In addition, they conclude that the variations in retreat rate between the ICESat and CryoSat-2 era are due to inter-decadal changes in ocean forcing.

The paper is well written and (as the authors state) provide much needed insights on a region of West Antarctica that has previously suffered from limited observations. The

sectors increasing contribution to the total Antarctic mass loss make these findings particularly relevant to the community and therefore appropriate for publication in TC. I believe however there are some points that need addressing, particularly in the discussion of their findings before it can be accepted for publication. These are outlined below.

General Comments

The title and the main arguments in the paper suggest the primary driver of change in the region is due to an inter-decadal climate-ocean variability. Whilst evidence of this is provided in the text to support this, it seems to give the impression that it is the sole dominant driver of change in the region. The text also states other factors such as geological controls and the effect of increased basal melt from neighbouring ice shelves on modifying the CDW. Unless it can be quantitatively proved that the impacts of these are minimal compared to the climate-ocean variability, then I think the emphasis placed on this sole factor needs to be better balanced with other potential drivers. I would also suggest revising the title to reflect this.

In the method section detailing the Swath SARIn processing of CryoSat-2 data and subsequent dh/dt calculations, it is stated that the plane fit following McMillan et al (2014) was used (P5, L18). This plane fitting approach was applied to POCA data and therefore includes a coefficient to account for firn penetration of the altimeter from ascending and descending passes (Supplementary material equation 1, McMillan et al 2014). As the Swath SARIn processing chain differs, is this approach still applicable? If the plane fit equation used in this instance differs from that in the McMillan paper, then it should be included in the main text (or as part of the supplementary information).

Technical Corrections

P1 L26 - "Projecting contributions" I would rephrase this to something like "accurately projecting the contribution of the West Antarctic Ice Sheet to global sea level rise".

P1, L25 - An extra reference should be added here as ice mass losses over the ASE have been measured from multiple techniques, not just the mass budget/IOM method. I would recommend adding Sutterley et al (2014), which shows mass losses over the region from separate techniques since 1992.

P1 L29 & L30 - Would it not be appropriate to put the Paolo et al (2015) reference with those regarding ice shelf melting as opposed to inland dynamic thinning?

P3, L17 - "Final Ib products were smoothed using standard GIS tools" - The specific tool or method should be stated for reproducibility purposes. Do the smoothing processes used change the position of the grounding line? Or is the extent of movement caused by this tool below the resolution for which the grounding line can be detected from Landsat 7/8 or ASTER?

P 10 L29 - P11 L2 - Is there any way to quantify this or explore this in more detail? As this implies that whilst the ocean-climate forcing is a major driver, at the local scale other factors could play a governing role in the rate of grounding line change.

P11 L13-15 - A recently published paper by Paolo et al (2018) looks at the impact of ENSO forcing on the West Antarctic Ice Shelves and seems to support your suggestions, so may be useful to add a reference to it here.

P15 L15-20 - This seems to suggest that inter-decadal climate-ocean variability is perhaps not the sole driver of change in this region. Is it possible to expand this discussion or quantify this effect? Otherwise a change of emphasis may be necessary to encompass these varying effects (see general comments above).

P15 L28-L29 - I don't think this statement can be made without quantitative analysis of other factors (as discussed above). I think this should be reworded to encompass this explanation is part of variety of factors affecting the Getz (particularly at the local scale).

P17 L10 - I would rephrase "RACMO2 and IMAU-FDM models used in the CryoSat-2

swath processing chain" as the models are not used in the generation of the swath data itself, but in determining ice shelf dh/dt.

P19 L15 - Fretwell et al reference does not list all paper authors

P23 L11 - Shepherd et al reference does not list all paper authors

Figure 1 - The scale bar in the bottom right hand corner is difficult to read against the background colour scheme, I would suggest changing it another colour (perhaps white).

Figure S1 - A scale bar should be added to this plot. In addition, I would suggest changing the ice shelf front position colour/line to make it more prominent compared to the grounding line delineation.

Figure S5 - The jet colour bar on this figure should be changed to avoid readability issues. This colour bar also clashes with the contour lines, making them difficult to view. The contour lines however could be kept as is, depending on choice of new colour table. Additionally, the colour bar needs to have a label stating what it is representing and the units.

---

## Referee Comment (RC3) · Anonymous Referee #3 · 10 Apr 2018

In this manuscript Christie and colleagues compare a newly developed product of grounding-line migration along the Marie Byrd Land sector with changes in surface elevation and discuss possible ocean forcing of the observed changes.

My limited knowledge doesn't allow me to comment on the quality of these products or on the method used to obtain them. However, I enjoyed reading the in depth analysis of possible ocean forcing and its mechanism. The authors computed the wind stress anomalies and the Ekman upwelling, looked at the configuration of the bottom topography, the location of the ACC and Antarctic slope current. This results in a very interesting investigation. However I think analysis could be made clearer. See here three examples to help doing it:

- The issue of why 33% of the grounding line retreated over the full 2003-2015 period

is not clear. This is reflected by the short and speculative section 4.1. I do not think this problem should be solved in this manuscript but a clear acknowledgment of the remaining unknowns seem necessary. Maybe stating clearly that the observation of long term grounding line retreat are probably linked to ocean forcing but this cannot be shown given the limited data available and the precise mechanism is unknown.

- In 4.2.1 the fact that grounding line retreat slew down during the 2010-2015 period but the shelf continued to thin is discussed. At the end different hypothesis are made to explain this apparent contradiction: "Possible confounding factors are surface mass balance processes, grounding zone bed geometry, and local-scale changes in ice dynamics. " These factors are discussed at length in the followinf sections so it would be interesting to have the answer to that contradiction in the conclusion.

- In 4.2.3 the authors say: "Until more comprehensive knowledge of Getz Ice Shelf's grounding zone bed structure exists, glacier/ice-stream- specific internal variability, moderated by bed conditions at the 2010-2015 grounding zone, cannot be reliably dismissed as an additional control on the slowdown of GL retreat rate during the CryoSat-2 era. " but then in the conclusion I read "We attribute the observed slowdown in Getz Ice Shelf's grounding-line retreat to a reduction in external climate-ocean forcing as inferred from climate reanalysis data. " This sounds contradictory, bed geometry might have played a role but the authors conclusion is still that ocean forcing is responsible for the slow down.

Minor comments:

The expression "climate-ocean" is strange, the ocean is part of the climate system, I think in most places it could be replaced by atmosphere-ocean or more explicitly "wind driven ocean".

p.1 l.15: "33% of the grounding line underwent retreat", how much underwent advance? It could help the reader by stating this here as well. I first thought this meant that 67% underwent advance.

p.6, l.28: f is the Coriolis parameter not the variations in the Coriolis parameter

---

## Author Comment (AC1) · 11 May 2018

**Marie Byrd Land glacier change driven by inter-decadal climate-ocean variability:**
**Author response to reviews**

Dear Dr. Wouters (Editor),

We thank all three reviewers for their insightful comments and feedback which have helped us to clarify our manuscript. In the following response document, we have compiled and numbered each of the reviewer's comments (*blue italics*), and include our response (black text) and amendments to the original text (*grey italics*). Page/line numbers refer to the original manuscript published on 29[th] January 2018 at: *https://www.the-cryosphere-discuss.net/tc-2017-263/tc-2017-263.pdf*.

We hope that you will find our amendments to the manuscript satisfactory for publication in TC, and we look forward to hearing from you soon.

Kind Regards,

Frazer
(on behalf of all co-authors)

**Reviewer 1**

1. *"The authors analyzed the glacier changes in Marie Byrd Land sector for the past 20 years. Using ICESat and CryoSat-2, they show that grounding line retreat reduced by 68% in CryoSat-2 era, which is caused by oceanic forcing. Although slowdown of grounding line retreat is an interesting and important finding, their argument that observed changes are caused by "reduced Ekman upwelling on and around the continental shelf" is not well supported. I recommend major revision. Very nice results, but some interpretations seem to me rather too speculative. More analyses and/or different interpretations are required".*

We are grateful to the reviewer for his/her interest in our work, and address his/her concerns regarding the need for a major revision in the following section.

**Major comment:**

2. *"Very good job listing oceanic processes, which may impact the glacier retreat. Although authors are aware of many oceanic processes, authors conclude that "during weaker offshore winds relative to the ICESat era reduced Ekman upwelling on and around the continental shelf, resulting in a decline in Circumpolar Deep Water intrusion to the sub Getz ice-shelf cavity", which is not supported from any of the analyses conducted in the paper".*

   *"As authors are aware, oceanic conditions (e.g., large and small scale circulations, bathymetry, stratification, etc) are very much different in the Marie Byrd Land sector. Since there are many processes potentially controlling potential temperature in the ice shelf cavity and thus ice shelf melt rates and relative importance of these processes are likely regionally different, authors are not able to conclude that "reduced Ekman upwelling on and around the continental shelf" is the key process for this region, just based on the fact that they observe changes in Ekman upwelling. Cited papers such as Steig et al., 2012, Dutrieux 2014, and St. Laurent et al., 2016 have conducted data analysis and/or modeling. Further analysis including data analysis and modeling is likely required to claim that "reduced Ekman upwelling on and around the continental shelf" is the reason for the observed changes".*

As the reviewer suggests, this study builds upon the work of -in particular- Thoma et al. (2008), Steig et al. (2012), Jacobs et al. (2013) and Dutrieux et al. (2014), who used a combination of ocean modelling (adapted to include sub-ice-shelf cavity bathymetry after Holland and Jenkins (2001); Thoma, Steig), atmospheric reanalysis data (Thoma, Steig, Dutrieux, Jacobs) and/or in-situ ocean observations (Dutrieux, Jacobs) to examine inter-annual-scale changes in oceanic forcing of the glaciers draining the Marie Byrd Land (Getz) and Amundsen Sectors.

In this contribution, we have reported on temporal changes in 10 m zonal wind and Ekman vertical velocities near to the continental-shelf break between 2003 and 2015, using virtually identical calculations to those employed by Steig/Dutrieux/Jacobs, and more recently by Greene et al. (2017) and Walker et al. (2017). These calculations have previously been shown to be highly correlated to both observed and modelled changes in the hydrography of the sub-shelf cavity (Dutrieux, Jacobs), thus we believe our estimates provide a reliable first-order proxy for the state of the ocean underneath Getz Ice Shelf between 2003 and 2015.

Whilst we fully agree that the local-to-large-scale processes controlling sub-cavity CDW availability are undoubtedly more complex than those captured by these simple diagnostic calculations (see the discussion in Section 4.2.1; also Webber et al. (2017) referenced therein), it is important to note that the ability to carry out in-situ-derived data analyses or

observationally-constrained ocean-modelling experiments over our study domain is currently limited owing to an almost complete dearth of either spatially or temporally continuous oceanic data during our observational window. To our knowledge, the most continuous spatial-temporal CTD observations along the length of Getz Ice Shelf were last acquired in 2000 and 2007, as reported in Jacobs et al. (2013). Jacobs et al. (2013) document a much increased oceanic forcing and observed melt rate of Getz in 2007, believed to be driven by an enhanced Ekman transport relative to 1999/2000 (i.e. reduced upwelling/CDW presence in 1999/2000, increased upwelling/CDW presence in 2007). Within the neighbouring and much more densely surveyed Amundsen Sea Embayment, Dutrieux et al. (2014) attributed the dramatic cooling of Pine Island Bay in recent years to a marked suppression of zonal wind stress (and by implication, Ekman upwelling) at the continental-shelf break around ~2011-2012. Resulting in a much-reduced thermocline and on-shelf CDW presence until at least 2014 (Webber et al., 2017), these changes and those of Jacobs et al. (2013) are fully consistent with our 1979-2017 time-series of *wE* shown in Figure 4 (b), which we believe act as important independent verifications of our MBLS calculations.

To further support our use of zonal wind/*wE* in the manuscript, we examined changes in the vertical hydrography of Getz' continental shelf and break, using the Met Office EN4 objective analysis solution for 2000-2017 (Figure 1 of this response document). Unlike ocean models or reanalysis data, this product is a monthly gridded interpolation of all available in-situ ocean observations derived from the World Ocean Database (WOD09/13) and other data sources (see Good et al. (2013) for further information). Whilst subject to high uncertainty bounds and coarse (1°) spatial resolution (Good et al., 2013), this result shows a clear reduction in thermocline depth between 2010-2015 relative to 2003-2008, with unprecedented deepening between 2013 and 2015. Following Dutrieux et al. (2014), a less significant yet clear decrease in thermocline depth is also witnessed c.2011-2012, as is a shallower thermocline in 2007 compared to year 2000 in accordance with the findings of Jacobs et al. (2013). These patterns are all consistent with the patterns of change shown in our manuscript's Figure 4 (b), and hence bolster our confidence in the ability of *wE* to act as a reliable indicator for the changing oceanic conditions near and underneath Getz Ice Shelf during 2003-2015.

[Figure]

**Figure 1**: Met Office EN4 objective analyses (potential temperature; °C) of the Getz region, derived from all model grid cells contained on the shelf and shelf break. Semi-transparent blue and red hatches denote the ICESat (2003-2008) and CryoSat-2 (2010-2015) observational periods (cf. Figure 4; main manuscript); thick black line denotes the +1°C isotherm ≈ the limits of the mCDW/CDW layer (cf. Jacobs et al., 2013). Dashed black line signifies the -300 m depth contour for reference.

There is one caveat to this EN4 analysis: the dense observational datasets acquired over the Amundsen Sea Embayment and Getz regions in recent years (e.g. Wåhlin et al., 2010; Jacobs et al., 2013; Dutrieux et al., 2014; Webber et al., 2017) may not be fully assimilated into this dataset. For this reason we present the EN4 analysis only in this response, rather than proposing its inclusion in the main text.

**Author amendments to manuscript:**
In summary of the above, we believe we are justified to use wind/Ekman anomalies in the manuscript as proxies for oceanic forcing of changes to the hydrography of Getz' sub-shelf cavity through time. Except for the coarse EN4 interpolation undertaken for this response (Figure 1), the dearth of continuous in-situ data collected near Getz Ice Shelf over the study period of 2003-2015 preclude further meaningful data analyses or modelling that the reviewer advocates.

That said, we fully agree with the reviewer that our calculations are a simplification of the undoubtedly complex range of processes controlling Getz' hydrography, and have clarified this point and justified our chosen methodology throughout the manuscript. Specifically:

i)      The abstract has been rewritten to emphasise the importance of bed topography and the more complex ice-sheet ocean interactions likely at work. (Latter section of abstract now reads: "*Along Getz Ice Shelf, grounding-line retreat reduced by 68% during the CryoSat-2 era relative to earlier observations. Climate reanalysis data reveal that wind-driven upwelling of Circumpolar Deep Water would have been reduced during this later period, suggesting that the observed slowdown was a response to reduced oceanic forcing. However, lack of comprehensive oceanographic and bathymetric information proximal to Getz Ice Shelf's grounding zone make it difficult to assess the role of intrinsic glacier dynamics, or more complex ice-sheet-ocean interactions, in moderating this slowdown. Collectively, our findings underscore the importance of spatial and inter-decadal variability in atmosphere and ocean interactions in moderating glaciological change around Antarctica*").

ii)     Section 2.3 has been modified to justify our choice of methods owing to the lack of observational data acquired over the MBLS during the 2003-2015 period
(Paragraph now reads: "*To investigate the role of atmospheric and oceanic forcing on glaciological change between 2003 and 2015, we examined mean zonal wind and Ekman vertical velocity anomalies on and near the MBLS' continental shelf, using ECMWF ERA-Interim climate reanalysis data (cf. Dee et al., 2011). These methods were utilised due to a dearth of spatially and temporally continuous in-situ oceanographical observations within the MBLS during the observational period, with the last comprehensive and publically-available surveys having been carried out in 2000 and 2007 (Jacobs et al., 2013)*").

iii)    Section 2.3.2. (Page 6, Line 19) has been reworded to re-emphasise that these calculations offer a first-order proxy for changes in Ekman transport-induced upwelling onto the continental shelf.
(Paragraph now reads: "*A derivative of the wind stress field, Ekman vertical velocity, approximates the rate at which the wind stress curl raises subsurface isopycnals, and can be used as a first-order estimate for Ekman transport-induced upwelling of interior ocean water masses, including relatively warm upper CDW layers (Marshall & Plumb, 2008)*").

iv)     The discussion in 4.2.1 (Page 12, Line 5-6) has been extended to assert the importance of acquiring observational data at/near Getz Ice Shelf in the coming years, to improve our understanding of the processes controlling change within Getz' sub-shelf cavity.
(Section now reads: *"Our findings underscore the potential importance of inter-decadal variability in both regional- and local-scale atmosphere and ocean interactions in moderating glaciological change along this sector of Antarctica (Fig. 6). Our observations also highlight the need for continuous in-situ ocean observations near and underneath Getz Ice Shelf in the future. Such observations would yield greater insight into the specific oceanographic mechanisms controlling the hydrography of Getz Ice Shelf's sub-shelf cavity (cf. Jacobs et al., 2013; Kim et al, 2017; Webber et al., 2017), beyond the approximations presented here and the spatially and temporally-limited observations previously reported (e.g. Wåhlin et al., 2010; Jacobs et al., 2013)…"*.

v)      In relation to i)-iv) above, the manuscript's conclusion has been reworded to reemphasise the complexity of the ocean interactions causing changes to the Getz sub-shelf cavity, and includes an explicit statement on the requirement for continuous ocean survey in the future (Page 15 Lines 21 to Page 16 Line 11).

Specifically, paragraph 2 now reads: "*We find a correspondence between the observed slowdown in Getz Ice Shelf's grounding-line retreat and a reduction in external atmosphere-ocean forcing as inferred from climate reanalysis data. During the CryoSat-2 era, weaker offshore winds relative to the ICESat era reduced Ekman upwelling on and around the continental shelf, resulting in a likely decline in Circumpolar Deep Water intrusion to the sub-Getz ice-shelf cavity. This is analogous to observed changes elsewhere in the Amundsen Sea Sector since 2009 (Dutrieux et al., 2014, following Steig et al., 2012; Turner et al., 2017), and is supported by empirically-constrained trends of oceanographic change observed near Getz Ice Shelf's calving fronts in the years immediately preceding the ICESat era (Jacobs et al., 2013). However, at the local scale, grounding zone bed geometry, which is poorly constrained along much of Getz Ice Shelf, may have also played a role in modulating retreat rates. Additional near-shore ocean processes, such as eddy-mediated transport of CDW across the shelf break (Stewart & Thompson, 2015), seasonal variations in on-shelf heat transport linked to local-scale atmospheric forcing (Webber et al., 2017) and the influence of sea ice on Ekman vertical velocities (Kim et al., 2017), may also have contributed to the observed reduction in GL retreat*".

Paragraph 4 now reads: "*Collectively, our findings from the Marie Byrd Land Sector underscore the importance of both spatial and inter-decadal variability in ocean and atmosphere interactions for moderating glaciological change around Antarctica. To assess the importance of these interactions, increased spatial-temporal oceanographical observations and high-resolution geophysical measurements of the MBLS' geological setting are required*".

3.  *The title of this paper indicates that glacier change is driven by inter-decadal climate ocean variability, which is misleading. Authors do not show that the impact of other processes are small. There are other processes impacting glacier retreat (section 4.2) and these processes may possibly be more important (e.g., subsection 4.2.3)".*

The reviewer acknowledges that we have detailed other possible influences on MBLS glacial change in our paper. To acknowledge the concern that the paper's title conveys

too much certainty that the observed glacial changes are driven exclusively from the atmosphere/ocean, we have amended the manuscript title to the following, which we believe also addresses the comments from Reviewers #2 (Comment 1) and #3 (Comments 2 and 3).

"*Glacier change along West Antarctica's Marie Byrd Land Sector and links to inter-decadal atmosphere-ocean variability*".

**Minor comments**

4. *"Page 9 Lines 14-25: It is clearer if authors can show spatial pattern of vertical Ekman velocity for each era (not just the difference as in Figure 5)".*

   Figure 5 now shows three subplots showing *wE* for (Jan) 2003 to (Jan) 2008 (a), (Jan) 2010 to (Dec) 2013 (b), and their difference (c). The figure caption has been reworded to reflect this change, along with several references to the new subplots in Sections 3.4 (Page 9 Lines 26-31) and 4.2.1 (Page 11 Line 26; Page 12 Line 3).

5. *"Page 11 Lines 10-11: Even if it is fully synchronous, it is not convincing that "reduced Ekman upwelling on and around the continental shelf" changes the oceanic condition in the ice shelf cavity, reduces the melt rates, and slows down the grounding line retreat. As stated above, there are many processes and further analyses are required".*

   We interpret this as essentially the same comment to which we have responded to the reviewer's comment 1 above.

6. *"Page 11 Lines 23-27: Where do you mean? Is there different polynya in near Getz region? If so, are these responding similarly to the Amundsen Sea polynya?"*

   This statement refers to the area of deep downwelling centred immediately north of Dotson Ice Shelf's calving front at ~115° W, which extends over the eastern and central tributaries of the Getz-Dotson Trough as seen in Figure 5. This area resides over the Amundsen Sector's largest polynya, commonly referred to in the literature as the 'Amundsen Sea Polynya' (cf. Nihashi & Ohshima, 2015; 2017; Kim et al., 2017).

7. *"Page 14 Lines 10-29: "These longitudinal limits corresponds broadly with . . . Getz Ice Shelf". These argument seems speculative. Need more clarification".*

   We are unsure why the reviewer finds this statement speculative, as the mean slope front identified by Whitworth et al. (1998) and Lee & Coward (2003) commences westward towards the Ross Sea at ~120° W and ~125-135° W, respectively, which is generally consistent with the regionally-contrasting glaciological behaviour (both GL retreat and ice surface elevation change rates) we observe at and west of Getz Ice Shelf. Representing a non-stationary, semi-permanent feature, the precise longitudinal limits of the ASF wax and wane and are controlled by a multitude of oceanic processes (cf. Jacobs et al., 1991; Baines et al., 2009).

8. *"Page 15 Lines 22-Page 16 Line 12: See major comments #1 and #2".*

   This comment refers to the reviewer's comments 2 and 3 according to our notation, which we have addressed above.

9. *"Figure 6: This seems to be not accurate. Should circulation off the Marie Byrd Land sector be influenced by the Ross Gyre and CDW circulation be opposite?"*
For simplicity, we originally did not include the influence of the Ross Gyre in Figure 6 owing to its non-stationary position through time. However, as the reviewer suggests, the presence of the Ross Gyre, which may intermittently encroach as far east as the Amundsen and Bellingshausen Sea regions (Assmann & Timmerman, 2005; Jacobs et al., 2013), may indeed influence the circulation of CDW along the MBLS coast. Therefore, we have decided to make reference to this phenomenon in the updated version of the manuscript.

To examine the influence of the Ross Gyre on the Marie Byrd Land Sector, we examined the vertical hydrography of the MBLS' coastline using the Met Office EN4 objective analysis product detailed in our response to comment '2' above. This was done by examining annually-averaged potential temperatures along the region west of 135° W (Figure 2), in comparison to the Getz–only region shown in Figure 1. Relative to Figure 1 (Getz), Figure 2 shows high temporal variability in 1°C thermocline depth and thickness of the warm (>1°C) ocean layer, in addition to a notable deepening of the shallower, colder (<0.5°C) waters. Consistent with vertical mixing or other oceanic processes associated with periodic incursions of the Ross Gyre, such an incursion may partly explain the deep downwelling centred at ~140° W during 2010-2015 (new Fig. 5b and c in main text), and implies a maximum easternmost boundary of ~129° W over our observational period.

[Figure]

**Figure 2**. Same as Figure 1 but for the region west of 135° W, derived from all model grid cells contained on the shelf and shelf break. Semi-transparent blue and red hatches denote the ICESat (2003-2008) and CryoSat-2 (2010-2015) observational periods (cf. Figure 4; main manuscript); thick black line denotes the +1°C isotherm ≈ the limits of the mCDW/CDW layer (cf. Jacobs et al., 2013). Dashed black line signifies the -300 m depth contour for reference.

To reflect the above, we have adapted Figure 6 of our manuscript to include the influence of the Ross Gyre in the region west of Getz (Figure 6 (b)). As the reviewer suggests, this now shows a reversal and cooling and/or downwelling of the ACC-derived CDW layer towards the Ross Sea.

In addition to this amendment, the following edits have been made to the main text and caption of Figure 6:

Page 15 Line 1: Sentence now reads: *"Influenced by the position of the Ross Gyre (Assmann & Timmerman, 2005), this ACC behaviour persists…."*.

Page 34 Line 3: Sentence now reads: *"The northward deflection of the ACC, influenced by the easternmost limits of the Ross Gyre, also minimises the presence of CDW near the shelf slope"*.

*"The authors present an analysis of changes in grounding line position for the Marie Byrd Land Sector using a well established method of detecting the break in slope from satellite remote sensing imagery (Landsat & ASTER). Over the 2003 - 2015 period they find that 33% of the grounding line underwent retreat, with the greatest rates found over the Getz Ice Shelf grounding line region. These results are consistent with the ice shelf thinning rates presented in the paper and previous studies in the literature. In addition, they conclude that the variations in retreat rate between the ICESat and CryoSat-2 era are due to inter-decadal changes in ocean forcing.*

*The paper is well written and (as the authors state) provide much needed insights on a region of West Antarctica that has previously suffered from limited observations. The sectors increasing contribution to the total Antarctic mass loss make these findings particularly relevant to the community and therefore appropriate for publication in TC. I believe however there are some points that need addressing, particularly in the discussion of their findings before it can be accepted for publication. These are outlined below."*

We are grateful to the reviewer for his/her thorough review of our manuscript and his/her insights.

**General Comments**

1. *"The title and the main arguments in the paper suggest the primary driver of change in the region is due to an inter-decadal climate-ocean variability. Whilst evidence of this is provided in the text to support this, it seems to give the impression that it is the sole dominant driver of change in the region. The text also states other factors such as geological controls and the effect of increased basal melt from neighbouring ice shelves on modifying the CDW. Unless it can be quantitatively proved that the impacts of these are minimal compared to the climate-ocean variability, then I think the emphasis placed on this sole factor needs to be better balanced with other potential drivers. I would also suggest revising the title to reflect this".*

   This comment echoes that of Reviewer# 1 and our actions in response are detailed in our response to Reviewer #1's comment 3) above.

2. *"In the method section detailing the Swath SARIn processing of CryoSat-2 data and subsequent dh/dt calculations, it is stated that the plane fit following McMillan et al (2014) was used (P5, L18). This plane fitting approach was applied to POCA data and therefore includes a coefficient to account for firn penetration of the altimeter from ascending and descending passes (Supplementary material equation 1, McMillan et al 2014). As the Swath SARIn processing chain differs, is this approach still applicable? If the plane fit equation used in this instance differs from that in the McMillan paper, then it should be included in the main text (or as part of the supplementary information)".*

   The reference to McMillan et al. (2014) was intended as a general reference to the plane fit approach as opposed to cross-over or repeat track techniques. However, we understand that this may lead to ambiguity as this specific aspect of the 'McMillan' inversion was not applied to our dataset (since swath data have an inherently different power structure to that of Point of Closest Approach (POCA) returns). To avoid this ambiguity, we now refer to Gourmelen et al. (2017b) when describing the plane fit solution on Page 5 Line 18. Sentence now reads: "*We derived linear rates of surface elevation change from time-dependent swath elevation data acquired between 2010 and 2016 using a plane fit approach on a 10 km grid posting (cf. Gourmelen et al., 2017b)*".

We have additionally amended all reference to Gourmelen et al. (2017) to "*Gourmelen et al. (2017a)*" in light of this amendment, and added Gourmelen et al. (2017b) to the reference list.

**Technical Corrections**

3. *"P1 L26 - "Projecting contributions" I would rephrase this to something like "accurately projecting the contribution of the West Antarctic Ice Sheet to global sea level rise"".*

   Page 1 Line 26: Sentence now reads: "*Comprehending the drivers of these ice losses is imperative for accurately projecting the contribution of the West Antarctic Ice Sheet to global sea level rise in the coming decades (e.g., Vaughan et al., 2013)*".

4. *"P1, L25 - An extra reference should be added here as ice mass losses over the ASE have been measured from multiple techniques, not just the mass budget/IOM method. I would recommend adding Sutterley et al (2014), which shows mass losses over the region from separate techniques since 1992".*

   Reference added to text and reference list.

5. *"P1 L29 & L30 - Would it not be appropriate to put the Paolo et al (2015) reference with those regarding ice shelf melting as opposed to inland dynamic thinning?"*

   Yes – thanks for pointing out our error here.

6. *"P3, L17 - "Final Ib products were smoothed using standard GIS tools" - The specific tool or method should be stated for reproducibility purposes. Do the smoothing processes used change the position of the grounding line? Or is the extent of movement caused by this tool below the resolution for which the grounding line can be detected from Landsat 7/8 or ASTER?"*

   We used a Polynomial Approximation with Exponential Kernel (PAEK) algorithm to smooth the $I_b$ products under a user-defined forward-looking tolerance limit, following Depoorter et al. (2013). For clarity here, a tolerance threshold of 500 m was chosen to smooth our $I_b$ products, which resulted in minimal changes in the position of the picked GL (generally at the sub-pixel scale). As the reviewer surmised, these modifications lie well below the resolution in which the GL could be detected from Landsat/ASTER, so are assumed to be negligible. This procedure has been detailed in Depoorter et al. (2013) as well as in the recently published technical metadata of Christie et al. (2018), which we have added to the paper's reference list and data availability section. We have therefore restructured the sentence to include these two references. Sentence now reads: "*The final $I_b$ products were smoothed using standard GIS tools (cf. Depoorter et al., 2013; Christie et al., 2018), and reflect the mean summertime GL position for each year as resolved from all available Landsat or ASTER imagery*".

7. *"P 10 L29 - P11 L2 - Is there any way to quantify this or explore this in more detail? As this implies that whilst the ocean-climate forcing is a major driver, at the local scale other factors could play a governing role in the rate of grounding line change".*

   This section was simply intended to act as a transition to the following more detailed discussion rather than to act as standalone text. More thorough discussion of the

constraints on glaciological change along this part of the ice shelf is presented in Sections 4.2-4.24. To clarify this purpose, we have added reference to the immediately following discussion section in the main text. This section now reads: "*Collectively, these observations imply that local-scale ice-ocean processes or geological configurations underneath the most easterly portion of Getz Ice Shelf may render the region relatively immune to ocean-forced dynamic thinning and subsequent GL retreat. These considerations are discussed in further detail next (Section 4.2)*".

8. *"P11 L13-15 - A recently published paper by Paolo et al (2018) looks at the impact of ENSO forcing on the West Antarctic Ice Shelves and seems to support your suggestions, so may be useful to add a reference to it here".*

This paper was not published when we submitted this manuscript but it certainly does support our work. We have reworked the closing sentence of this paragraph to incorporate the citation, and added Paolo et al. (2018) to the reference list. Sentence (Page 11 Line 15-17) now reads: "*This hypothesis concurs with the recent findings of Paolo et al. (2018) who examined the response of ENSO variability on all Pacific-facing ice shelves over the radar altimetry record (1994-2017), as well as the earlier findings of Jacobs et al. (2013), who attributed a reduced thermocline and glaciological forcing along Getz Ice Shelf in the years preceding the ICESat era to a strong La Niña event circa. 2000*".

9. *"P15 L15-20 - This seems to suggest that inter-decadal climate-ocean variability is perhaps not the sole driver of change in this region. Is it possible to expand this discussion or quantify this effect? Otherwise a change of emphasis may be necessary to encompass these varying effects (see general comments above)".*

See our response to this Reviewer's Comment 1 and in turn our response to Reviewer #1 Comment 3. The point is well raised and accepted.

10. *"P15 L28-L29 - I don't think this statement can be made without quantitative analysis of other factors (as discussed above). I think this should be reworded to encompass this explanation is part of variety of factors affecting the Getz (particularly at the local scale)".*

Also addressed in our response to this Reviewer's Comment 1 and in turn our response to Reviewer #1 Comment 3 above.

11. *"P17 L10 - I would rephrase "RACMO2 and IMAU-FDM models used in the CryoSat-2 swath processing chain" as the models are not used in the generation of the swath data itself, but in determining ice shelf dh/dt".*

Thanks – we have replaced the word "swath" here with "$\Delta h/\Delta t$".

12. *"P19 L15 - Fretwell et al reference does not list all paper authors".*
13. *"P23 L11 - Shepherd et al reference does not list all paper authors".*

Both changed to include full author lists.

14. *"Figure 1 - The scale bar in the bottom right hand corner is difficult to read against the background colour scheme, I would suggest changing it another colour (perhaps white)".*

Scale bar changed to white.

15. *"Figure S1 - A scale bar should be added to this plot. In addition, I would suggest changing the ice shelf front position colour/line to make it more prominent compared to the grounding line delineation".*

Scale bar added, and we have changed the colour and thickness of the ice shelf fronts to add greater contrast against the delineated GLs.

16. *"Figure S5 - The jet colour bar on this figure should be changed to avoid readability issues. This colour bar also clashes with the contour lines, making them difficult to view. The contour lines however could be kept as is, depending on choice of new colour table. Additionally, the colour bar needs to have a label stating what it is representing and the units".*

We have redesigned Figure S5 to incorporate a new colour map as the reviewer suggested, and have added in the colour legend. We have also changed the colour of the contours to ease readability.

*"In this manuscript Christie and colleagues compare a newly developed product of grounding-line migration along the Marie Byrd Land sector with changes in surface elevation and discuss possible ocean forcing of the observed changes. My limited knowledge doesn't allow me to comment on the quality of these products or on the method used to obtain them. However, I enjoyed reading the in depth analysis of possible ocean forcing and its mechanism. The authors computed the wind stress anomalies and the Ekman upwelling, looked at the configuration of the bottom topography, the location of the ACC and Antarctic slope current. This results in a very interesting investigation. However I think analysis could be made clearer. See here three examples to help doing it:"*

We thank the reviewer for his/her kind words and interest in our manuscript, and address his/her suggested revisions below.

1. *"The issue of why 33% of the grounding line retreated over the full 2003-2015 period is not clear. This is reflected by the short and speculative section 4.1. I do not think this problem should be solved in this manuscript but a clear acknowledgment of the remaining unknowns seem necessary. Maybe stating clearly that the observation of long term grounding line retreat are probably linked to ocean forcing but this cannot be shown given the limited data available and the precise mechanism is unknown".*

   Section 4.1 was merely intended to build the foundations for the immediately following sections, which provide detailed discussion on the ocean and other mechanisms driving change (or lack thereof) throughout the MBLS. We have added a note to the section to this effect (see response to Reviewer #2, comment 7).

   It is important to note that whilst GL migration and ice-shelf thinning rates abated during 2010-2015 relative to 2003-2008 (Figs. 2 and 3), the observed changes during this epoch were still almost exclusively defined by GL retreat and negative ice-shelf thickness change (implying thinning) rates. This indicates that throughout 2003-2015 the region has been in a state of dynamical imbalance, even if the retreat has reduced in the later period.

   To add a little more clarity on this point we have reworded part of Section 4.2, paragraph 1 to mention explicitly the role of dynamic imbalance throughout the observational period. The paragraph (Page 10 Line 15-25) now reads: *"The most prominent GL retreat throughout the MBLS occurred along the ~650 km Getz Ice Shelf (Fig. 1; Sect. 3.1), which neighbours the recent, rapidly downwasting ice masses of the wider Amundsen Sea Sector. This was a likely consequence of the substantial thinning and basal melting witnessed over this region in recent decades, indicative of an ongoing dynamically-driven glaciological imbalance through time (Figs. 2 and 3; see also Pritchard et al., 2009; 2012; Jacobs et al., 2013; Paolo et al., 2015). Indeed, all GL retreat within the central and western sectors of Getz Ice Shelf occurred directly upstream of well-surveyed, deep (>400 m) bathymetric depressions north of the ice fronts (Fig. 1) …"*.

2. *"In 4.2.1 the fact that grounding line retreat slew down during the 2010-2015 period but the shelf continued to thin is discussed. At the end different hypothesis are made to explain this apparent contradiction: "Possible confounding factors are surface mass balance processes, grounding zone bed geometry, and local-scale changes in ice dynamics". These factors are discussed at length in the followinf sections so it would be interesting to have the answer to that contradiction in the conclusion".*

This comment closely echoes those made by Reviewer #1 (comments 3 and 9) and Reviewer #2 (comments 1, 9 and 10), which we have explicitly addressed in our responses above. This has involved the reworking of Section 5 to provide a more balanced discussion of the processes controlling the observed changes across Getz Ice Shelf and the wider-MBLS.

3. *"In 4.2.3 the authors say: "Until more comprehensive knowledge of Getz Ice Shelf's grounding zone bed structure exists, glacier/ice-stream- specific internal variability, moderated by bed conditions at the 2010-2015 grounding zone, cannot be reliably dismissed as an additional control on the slowdown of GL retreat rate during the CryoSat-2 era. " but then in the conclusion I read "We attribute the observed slowdown in Getz Ice Shelf's grounding-line retreat to a reduction in external climate-ocean forcing as inferred from climate reanalysis data. " This sounds contradictory, bed geometry might have played a role but the authors conclusion is still that ocean forcing is responsible for the slow down".*

This is a further echo of Reviewer #1 comment 3 and Reviewer #2 comment 1, and has been actioned accordingly.

**Minor comments:**

4. *"The expression "climate-ocean" is strange, the ocean is part of the climate system, I think in most places it could be replaced by atmosphere-ocean or more explicitly "wind driven ocean"".*

All instances of "climate-ocean" in the manuscript have been changed to "atmosphere-ocean".

5. *"p.1 l.15: "33% of the grounding line underwent retreat", how much underwent advance? It could help the reader by stating this here as well. I first thought this meant that 67% underwent advance".*

We have extended this sentence (Page 1 Line 15) to read: "*During the observational period, 33% of the grounding line underwent retreat, with no significant advance recorded over the remainder of the ~2200 km long coastline*".

6. *"p.6, l.28: f is the Coriolis parameter not the variations in the Coriolis parameter".*

Yes - we have reworded this sentence to read: "*where f denotes variations in the Coriolis parameter at latitude φ; ω is the Earth's angular velocity (7.292 × 10-5 rad s-1); and $p_w$ is the density of the Ekman layer ocean water (1027.5 kg m-3)*".

**References stated in this response document, but not included in main text**

Good, S. A., Martin, M. J., and Rayner, N. A.: EN4: Quality controlled ocean temperature and salinity profiles and monthly objective analyses with uncertainty estimates, J. Geophys. Res. Oceans, 118, 6704–6716, doi:10.1002/2013JC009067, 2013.

Greene, C. A., Blankenship, D. D., Gwyther, D. E., Silvano, A., and van Wijk, E.: Wind causes Totten Ice Shelf melt and acceleration, Sci. Adv., 3(11), doi:10.1126/sciadv.1701681, 2017.

Holland, D. M., and Jenkins, A.: Adaptation of an isopycnic coordinate ocean model for the study of circulation beneath ice shelves, Mon. Weather Rev., 129(8), 1905–1927, https://doi.org/10.1175/1520-0493(2001)129<1905:AOAICO>2.0.CO;2, 2001.

Walker, C. C., and A.S. Gardner, A. S.: Rapid drawdown of Antarctica's Wordie Ice Shelf glaciers in response to ENSO/Southern Annular Mode-driven warming in the Southern Ocean, Earth Planet. Sci. Lett., 476, 100-110, doi:10.1016/j.epsl.2017.08.005, 2017.

---

## Author Response (AR2)

**Glacier change along West Antarctica's Marie Byrd Land Sector and links to inter-decadal atmosphere-ocean variability:**
**Author response to reviews**

Dear Dr. Wouters (Editor),

We thank the two reviewers for their post-revision comments, and are glad to see they are satisfied with our implemented revisions. In accordance with your editor report and the comments of reviewer #2, we have incorporated the following amendments to the manuscript as detailed below. As before, we have compiled and numbered each of the reviewer's comments (*blue italics*), and include our response (black text) and amendments to the original text (*grey italics*). Page/line numbers refer to the revised manuscript uploaded on 11[th] May 2018. We have retained our initial revisions to the manuscript in **red bold font** in the attached tracked-changed manuscript, and include our most recent revisions in **green bold font**.

We hope that you will find our amendments to the manuscript satisfactory for publication in TC, and we look forward to hearing from you soon.

Kind Regards,

Frazer
(on behalf of all co-authors)

**Reviewer #2 comments**

1. *"A minor revision I would recommend is the inclusion of Fig 1 from the author response file (perhaps as a panel in Figure 4). Whilst I agree with the authors regarding the interpolation caveats and the uncertainties regarding the data product containing the latest observations, I still think it is a useful analysis from observational data which supports the arguments made from the ERA-Interim model analysis. As long as the caveats are discussed in the main text I don't see a problem with its inclusion".*

Accepted. We have reworded the discussion of the EN4 dataset in the author response file and converted this into a new methods section (Section 2.3.3), which details the dataset and its main caveats. To support our arguments, we have also added several references to the observed changes in EN4-derived ocean temperature to the discussion (Section 4.2.1). Other, minor references pertaining to the dataset's inclusion have been added to Sections 2.3 and 5, and several associated citations have been added to the reference list. As the reviewer suggests, Fig.

1 has become a new panel in Fig. 4, and its figure caption has been updated to reflect this amendment.

Section 2.3. now reads: "*To investigate the role of atmospheric and oceanic forcing on glaciological change between 2003 and 2015, we examined mean zonal wind and Ekman vertical velocity anomalies on and near the MBLS' continental shelf, using ECMWF ERA-Interim climate reanalysis data (cf. Dee et al., 2011). To supplement our analyses, we also compared these datasets to changes in sub-surface ocean temperature derived from Met Office EN4 objective analysis products (cf. Good et al., 2013)...*".

Section 2.3.3 now reads: "*Changes in subsurface potential temperature linked to temporal variability in atmospheric forcing were examined using monthly Met Office EN4 objective analysis solutions for 2000-2017 (Good et al., 2013). Unlike ocean models or reanalysis data, these products represent quality-controlled, monthly gridded interpolations of all available in-situ ocean observations assimilated from the World Ocean Database (WOD09/13), the Global Temperature and Salinity Profile Program (GTSPP), and global Argo float data; Good et al. (2013) provide a thorough discussion of these data sources and their interpolation methodologies. While subject to high uncertainty (see Good et al. 2013 for further discussion) and coarse (1°x1°) spatial resolution, these observations provide an independent first-order impression of changes in the Southern Ocean's vertical hydrography (cf. Miles et al., 2016) to support our climate reanalysis records. We derived annually-averaged estimates of ocean potential temperature using EN4.2.1 data for all months between January 2000 and December 2017 (inclusive), and used the time-variable mechanical and expendable bathythermograph bias corrections of Gouretski and Reseghetti (2010) in our analyses*".

Section 4.2.1 now reads: "*Indeed, the changes in 10 m zonal wind, U (Fig. 4a), and Ekman vertical velocity, wE (Fig. 4b), suggest that the 2010-2015 era was characterised by a predominantly easterly wind anomaly over the MBLS CSB (Sect. 3.4), in conjunction with an implied regional-scale reduction in CDW upwelling and flooding onto the continental shelf. These findings are consistent with our EN4-derived observations of overall subsurface cooling near Getz Ice Shelf during 2010-2015 (Fig. 4c), and with similar, synchronous ocean-atmosphere trends observed over the wider Amundsen Sea Sector since 2009 which were responsible for a much deepened CDW layer across this region relative to the ICESat era (cf. Wåhlin et al., 2010; Jacobs et al., 2013; Dutrieux et al., 2014; Webber et al., 2017….*

*Likely related to changes in the spatial extent of the Amundsen Sea Polynya since 2003-2008 (cf. Nihashi & Ohshima, 2015; 2017; Kim et al., 2017), we hypothesise that the net reduced upwelling inferred to have occurred at this coastal location, consistent with a much deepened isopycnal depth over the trough's tributaries, may have been responsible for an additional, locally-forced reduction in central and eastern Getz-bound CDW inflow during 2010-2015. Echoed in our*

*EN4 observations of near-shore potential temperature change (Fig. 4c), such a reduction –
together with observations of deeply reduced wE along much of the remaining Getz Ice Shelf and
wider MBLS between 2010 and 2014 (Figs. 5c)- may explain the overall, central-western Getz-
wide reduction in GL retreat …".*

Section 5 now reads: "*We find a correspondence between the observed slowdown in Getz Ice
Shelf's grounding-line retreat and a reduction in external atmosphere-ocean forcing as inferred
from climate reanalysis and ocean objective analysis datasets*".

2. *P1, L19 "Climate reanalysis data reveal" - I think the word "reveal" should be changed to
something along the lines on "implies".*

Accepted. Now reads "*implies*'.

**Editor Comments**

1. "*I second R2's comment that it would be worthwhile to include one or both of the EN4 figures in
the final manuscript… I understand your concerns about the data, and these data should only be
used to support your arguments (rather than basing your arguments on the data), but as long as
the limitations are clearly explained, this shouldn't be a problem*".

Accepted as discussed above, with the exception of the inclusion of the second figure. Since the
submission of our revised manuscript, a new journal article examining the role of the Ross Gyre
and it's links to Antarctic Circumpolar Current has been published (Dotto et al., 2018). Detailing
the recent variability of the Ross Gyre as detected from satellite altimetry, this paper greatly
supports our arguments and offers insight into the implications of its presence/extent to nearby
ice-sheet melting rates beyond those presented in Fig. 2. Therefore, we have instead made
explicit reference to this paper on Pg. 15 Ln 14.

[revised manuscript text omitted]